EMBO
Molecular Medicine

# Modulation of mTOR signaling as a strategy for the treatment of Pompe disease

Jeong-A Lim[1,2,†,‡], Lishu Li[1,*,†,‡] [ID], Orian S Shirihai[3] [ID], Kyle M Trudeau[3], Rosa Puertollano[2,**,‡] [ID] & Nina Raben[1,***,‡] [ID]

## Abstract

Mechanistic target of rapamycin (mTOR) coordinates biosynthetic and catabolic processes in response to multiple extracellular and intracellular signals including growth factors and nutrients. This serine/threonine kinase has long been known as a critical regulator of muscle mass. The recent finding that the decision regarding its activation/inactivation takes place at the lysosome undeniably brings mTOR into the field of lysosomal storage diseases. In this study, we have examined the involvement of the mTOR pathway in the pathophysiology of a severe muscle wasting condition, Pompe disease, caused by excessive accumulation of lysosomal glycogen. Here, we report the dysregulation of mTOR signaling in the diseased muscle cells, and we focus on potential sites for therapeutic intervention. Reactivation of mTOR in the whole muscle of Pompe mice by TSC knockdown resulted in the reversal of atrophy and a striking removal of autophagic buildup. Of particular interest, we found that the aberrant mTOR signaling can be reversed by arginine. This finding can be translated into the clinic and may become a paradigm for targeted therapy in lysosomal, metabolic, and neuromuscular diseases.

**Keywords** autophagy; lysosomal storage disorders; mTOR; myopathy; Pompe disease

**Subject Categories** Genetics, Gene Therapy & Genetic Disease; Metabolism; Musculoskeletal System

## Introduction

Mechanistic target of rapamycin (mTOR), a highly conserved serine/threonine kinase, forms two multiprotein complexes, mTOR complex 1 (TORC1) and mTOR complex 2 (TORC2).

Rapamycin-sensitive mTORC1 complex responds to multiple signals, and when activated, changes the cell metabolism from catabolic to anabolic program, thus promoting protein synthesis and cell growth while repressing autophagy. The role of lysosome in controlling metabolic programs is emphasized by the discovery that activation of this potent anabolic regulator happens at the lysosome in a process mediated through an amino acid-sensing cascade involving V-ATPase, Ragulator, and Rag GTPases. When cells have sufficient amino acids, V-ATPase promotes the guanine nucleotide exchange factor (GEF) activity of Ragulator leading to the formation of active RagA/B·GTP complex at the lysosome; in this active configuration, Rag binds to and delivers mTORC1 to the lysosome where the kinase is activated by Rheb (Ras homolog enriched in brain), a small GTPase that is fixed to the lysosomal surface (Sancak *et al*, 2010; Zoncu *et al*, 2011; reviewed in Bar-Peled & Sabatini, 2014). Rheb is a downstream target of tuberous sclerosis complex (TSC) that functions as a GTPase-activating protein (GAP) and converts active GTP-bound Rheb to inactive GDP-bound form, thus inhibiting mTORC1 activity (Inoki *et al*, 2003; Demetriades *et al*, 2014; Menon *et al*, 2014).

The recent view of the lysosomes as a site of the mTORC1 activation, along with the long-established role of this kinase in the control of muscle mass, has made the study of mTORC1 signaling of particular interest to research on Pompe disease, a severe muscle wasting disorder characterized by altered lysosomal function. Profound muscle atrophy is a hallmark of Pompe disease, a rare genetic disorder caused by a deficiency of acid alpha-glucosidase (GAA), the enzyme that breaks down glycogen to glucose within lysosomes. Absence of the enzyme leads to a rapidly fatal cardiomyopathy and skeletal muscle myopathy in infants; low levels of residual enzyme activity are associated with childhood and adult-onset progressive skeletal muscle myopathy usually without cardiac involvement (Van der Ploeg & Reuser, 2008).

The introduction of enzyme replacement therapy (ERT) changed the natural course of the infantile form because of the notable effect in cardiac muscle; however, the effect in skeletal muscle has been

1   Laboratory of Muscle Stem Cells and Gene Regulation, National Institute of Arthritis and Musculoskeletal and Skin Diseases, National Institutes of Health, Bethesda, MD, USA
2   Cell Biology and Physiology Center, National Heart, Lung, and Blood Institute, National Institutes of Health, Bethesda, MD, USA
3   Department of Medicine, Obesity and Nutrition Section, Evans Biomedical Research Center, Boston University School of Medicine, Boston, MA, USA
    *Corresponding author. E-mail: lishuli1@hotmail.com
    **Corresponding author. Tel: +1 301 451 2361; E-mail: puertolr@mail.nih.gov
    ***Corresponding author. Tel: +1 301 496 1474; E-mail: rabenn@mail.nih.gov
    †These authors contributed equally to this work
    ‡This article has been contributed to by US Government employees and their work is in the public domain in the USA

modest at best (Kishnani et al, 2007; Strothotte et al, 2010; Van der Ploeg et al, 2010; Prater et al, 2012). The pathophysiology of muscle damage involves enlargement and rupture of glycogen-filled lysosomes, disturbance of calcium homeostasis and endocytic trafficking, mitochondrial abnormalities, and autophagic defect (Thurberg et al, 2006; Lim et al, 2014, 2015; Nascimbeni et al, 2015). The search for a more effective therapy is currently underway. However, even if muscles are cleared of glycogen and autophagic debris—the two major pathologies in Pompe disease—profound muscle wasting will persist and will remain a major therapeutic challenge.

The signaling pathways responsible for the loss of muscle mass in Pompe disease are largely unknown, and the reported studies on mTOR signaling yielded conflicting results. The temptation to boost protein synthesis by stimulating the mTOR pathway is reflected in recent data showing that leucine supplementation halted the decline in muscle mass and reduced glycogen accumulation in GAA-KO muscle (Shemesh et al, 2014). On the other hand, clearance of the excess muscle glycogen was reported following treatment of GAA-KO mice with mTORC1 inhibitor rapamycin; co-administration of rapamycin with the replacement enzyme (recombinant human GAA; alglucosidase alfa, Myozyme®, Genzyme Corporation, a Sanofi Company) reduced muscle glycogen content more than rhGAA or rapamycin alone (Ashe et al, 2010).

This study is the first systematic analysis of the upstream regulators and downstream targets of mTORC1 in Pompe muscle cells. We have found a dysregulation of mTOR signaling in the diseased cells —a diminished basal level of mTOR activity, weakened response to cellular stress, and the failure to reallocate mTOR away from lysosomes upon starvation. We have elucidated the molecular mechanisms underlying mTOR dysregulation in Pompe disease and identified points for therapeutic intervention along the mTOR signaling pathway. Furthermore, we have used targeted approaches to reverse the abnormalities in this prototypical lysosomal storage disorder.

# Results

### Perturbed mTOR signaling in cultured Pompe muscle cells

To explore the mTOR signaling pathway in Pompe disease, we took advantage of a recently developed in vitro model of the disease—GAA-deficient myotubes. These myotubes are formed from conditionally immortalized myoblasts derived from the GAA-KO mice; differentiated myotubes, but not myoblasts, contain large glycogen-filled lysosomes, thus replicating the disease phenotype (Spampanato et al, 2013). Since mTOR kinase is a principal regulator of protein synthesis, we evaluated the rate of protein synthesis in KO cells by using a surface sensing of translation (SUnSET) method, which relies on the incorporation of puromycin into nascent peptide chains resulting in the termination of their elongation (Goodman et al, 2011). A significant decrease (~60%) in anti-puromycin immunoreactivity was detected in KO myotubes compared to WT controls, a finding consistent with a reduction in protein translation (Fig 1A and B).

Phosphorylation of the 4E-BP1 repressor protein, a downstream mTOR target (Hay & Sonenberg, 2004) reduces its affinity for eIF4E

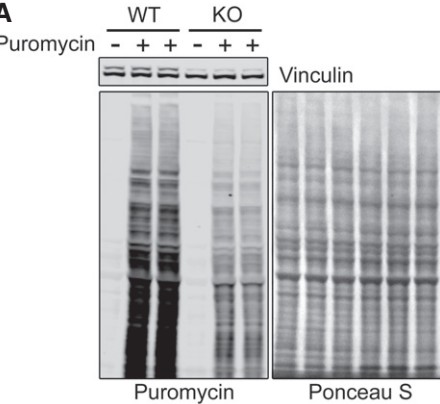

**A**

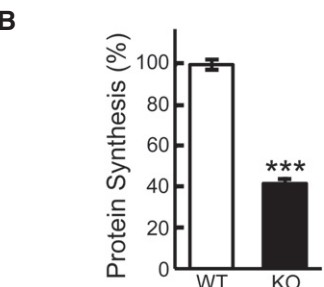

**B**

**Figure 1. Decreased protein translation in KO cells.**
Surface sensing of translation (SUnSET) analysis was used to evaluate the incorporation of puromycin into nascent polypeptides.

A   Representative image of Western blot analysis of WT and KO cells treated with puromycin (1 μM) for 30 min. Western blot with anti-vinculin antibody and Ponceau S staining were used as loading controls.
B   Total intensity of puromycin-labeled polypeptides was quantified. Student's t-test was used for statistical analysis. Data are mean ± SE. ***$P < 0.001$ ($P = 0.0009$; $n = 4$).

Source data are available online for this figure.

which can then associate with eIF4G to form active eIF4E·eIF4G complex, thus initiating cap-dependent translation. Unexpectedly, the level of phosphorylated 4E-BP1 (p-4E-BP1$^{T37/46}$) was significantly higher in KO myotubes compared to WT controls (Fig 2A) but the abundance of total 4E-BP1 followed a similar trend, thus confounding the assessment of mTORC1 involvement in protein translation. Therefore, we compared the abundance of eIF4E bound to 4E-BP1 in WT and KO cells; immunoprecipitation of 4E-BP1 from the WT and KO cell lysates followed by Western blotting with eIF4E (and vice versa) antibodies showed an increased eIF4E/4E-BP1 binding in KO cells (Fig 2B), suggesting insufficient mTORC1 activity and suppression of protein synthesis. Consistent with these data, the abundance of non-phosphorylated 4E-BP1 (active form), which binds to and reduces eIF4E availability, was increased in KO cells (Fig 2A).

A striking enhancement of 4E-BP1 translation despite the general inhibition of protein synthesis in KO cells prompted us to look at the eIF2α/ATF4 pathway. The phosphorylation of eIF2α represses global translation, but leads to increased translation of ATF4 (activation transcription factor 4), which regulates the transcription of many genes [reviewed in Sonenberg and Hinnebusch (2009)]; 4E-BP1 is a potential target of ATF4 because Eif4ebp1 gene contains

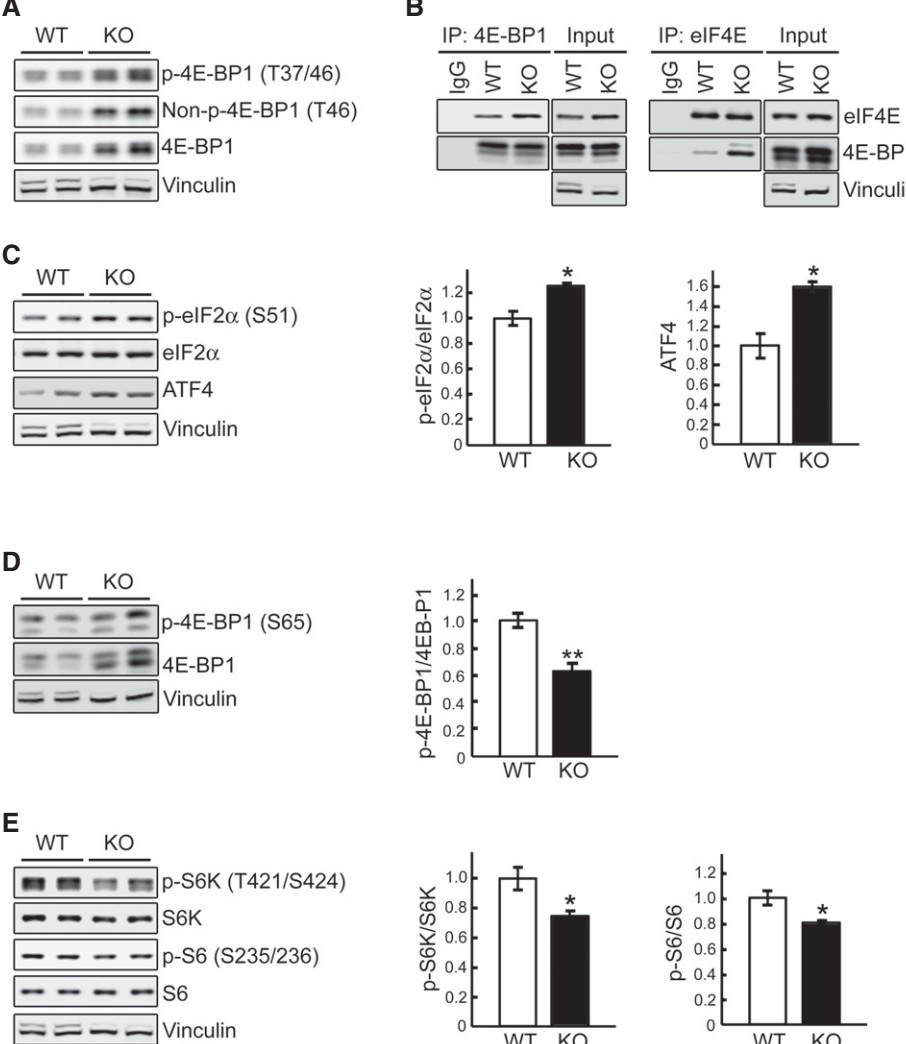

**Figure 2.  Dysregulation of mTOR signaling and activation of eIF2α/ATF4 pathway in KO cells.**

A   Representative Western blot of total lysates of WT and KO myotubes. All three forms of 4E-BP1, phosphorylated (p-4E-BP1$^{T37/46}$), non-phosphorylated (Non-p-4E-BP1$^{T46}$), and total, are increased in KO cells.

B   WT and KO cell lysates were immunoprecipitated (IP) with either anti-4E-BP1 (left) or anti-eIF4E (right); the immunoprecipitated proteins were then probed with eIF4E or 4E-BP1, respectively. Increased eIF4E/4E-BP1 binding is seen in KO cells. Immunoprecipitation with IgG was included as negative control.

C   Representative Western blot of total lysates of WT and KO myotubes with the indicated antibodies. The levels of eIF2α$^{S51}$ and ATF4 are increased in KO cells; graphs represent mean ± SE of p-eIF2α/eIF2α ratios ($n = 3$) and ATF4 ($n = 3$) levels. *$P < 0.05$, Student's *t*-test.

D   Immunoblot analysis of WT and KO lysates showing an increase in both total and p-4E-BP1$^{S65}$ and a decrease in p-4E-BP1$^{S65}$/4E-BP1 ratio in KO cells; graph represents mean ± SE ($n = 6$). **$P < 0.01$, Student's *t*-test.

E   Immunoblot analysis of WT and KO lysates showing a decrease in the p-S6K/S6K ($n = 5$) and p-S6/S6 ($n = 6$) ratios in KO cells; graphs represent mean ± SE. *$P < 0.05$, Student's *t*-test.

Data information: Vinculin was used as a loading control (vinculin and its splice variant are commonly seen in both WT and KO, although the ratio of these forms is different; both bands are used for quantitative analysis). All blots (except for IP) are representative of at least three independent experiments.
Source data are available online for this figure.

the ATF4-responsive elements (Kilberg *et al*, 2009). Indeed, we have found an increase in the levels of p-eIF2α$^{S51}$ and ATF4 in the KO (Fig 2C). eIF2α/ATF4 pathway plays an important role in the adaptation to stress caused by the generation of reactive oxygen species (Rajesh *et al*, 2015)—a condition observed in KO cells (Lim *et al*, 2015).

To better assess mTORC1 activity, we evaluated 4E-BP1$^{S65}$ phosphorylation which is a more reliable indicator since this site is

serum- and rapamycin-sensitive, whereas p-4E-BP1$^{T37/46}$ is only partially sensitive to these treatments (Gingras *et al*, 2001). Again, we found an increase in both forms, but the ratio of p-4E-BP1$^{S65}$/total was decreased in Pompe muscle cells, suggesting a diminished mTOR activity (Fig 2D). We then studied the phosphorylation state of S6K, a direct mTORC1 substrate that phosphorylates the ribosomal protein 6 (S6) of the 40S ribosomal subunit. The ratios of p-S6K$^{T421/S424}$/total S6K and p-S6$^{S235/236}$/total were decreased in the

KO cells (Fig 2E), again suggesting a compromised mTORC1 activity.

Next, we began analysis of the upstream inputs to mTORC1—the phosphorylation status of the major upstream regulators of mTOR, AKT, and the AMP-activated protein kinase (AMPK), which have opposite effect on mTORC1 activity. AKT activates mTORC1 through the inhibitory phosphorylation of the tuberous sclerosis complex 2 (TSC2$^{T1426}$); AMPK-mediated phosphorylation of TSC2$^{S1387}$ leads to its activation and suppression of mTORC1 (Inoki et al, 2003; Huang & Manning, 2008; Sengupta et al, 2010).

The phosphorylation levels of AKT (p-AKT$^{S473}$) were similar in WT and KO cells on days 4-5 in differentiation medium (not shown) and were even increased in the KO at a later stage of myotubes differentiation (Fig 3A). Despite this increase, AKT-mediated phosphorylation of TSC2$^{T1426}$ was decreased in KO cells, suggesting a failure of AKT to inhibit TSC2 (Fig 3A). On the other hand, the level of TSC2, as well as the levels of active phosphorylated form of AMPKα (p-AMPKα$^{T172}$) and its downstream target, p-ACC$^{S79}$, was increased in KO cell lysates (Fig 3A). AMPKα is activated under low-energy conditions (Sengupta et al, 2010), and its increased activity in KO cells under basal condition is not unexpected; the failure to digest lysosomal glycogen to glucose may deprive muscle cells of a source of energy (Fukuda et al, 2006).

Recent data have demonstrated that LKB1-mediated phosphorylation of AMPKα in response to energy stress takes place at the endosomal/lysosomal surface leading to inactivation of mTOR and its dissociation from endosome (Zhang et al, 2014). Therefore, we examined the levels of LKB1, AMPKα, and TSC2 in the lysosome-enriched fraction from KO cells (the purity of this fraction is shown in Appendix Fig S1). Indeed, the levels of all three proteins were elevated in KO cells compared to WT (Fig 3B). Furthermore, the levels of p-AMPKα$^{T172}$ and active phosphorylated form of TSC2$^{S1387}$ (AMPK-mediated phosphorylation) were increased in the lysosomal fraction in KO cells (Fig 3B). Since TSC2 inhibits mTOR by inactivating the small GTPase Rheb (Inoki et al, 2003), increased levels of AMPK activated TSC2 at the lysosomal surface may explain diminished basal mTOR activity in KO cells. Of note, AKT-mediated phosphorylated form of TSC2$^{T1426}$ was not detected in the lysosomal fraction (not shown).

### In vivo results mirror the findings in cultured cells

To validate in vivo the relevance of our in vitro findings, we analyzed mTOR signaling in whole muscle of the GAA-KO mice. For these studies, we have used the white part of the gastrocnemius muscle, which are most resistant to ERT (Lim et al, 2014). Significant portions of 4E-BP1 and S6 remained non-phosphorylated (decreased ratios of p-4E-BP1$^{T37/46}$/total 4E-BP1 and p-S6$^{S235/236}$/total S6) in GAA-KO muscle compared to WT, suggesting a decrease in mTORC1 activity (Fig 4A and B). Of note, similar to what was found in cultured KO cells, the levels of both p-4E-BP1 and total 4E-BP1 were increased in GAA-KO muscle, consistent with our previous data (Raben et al, 2010).

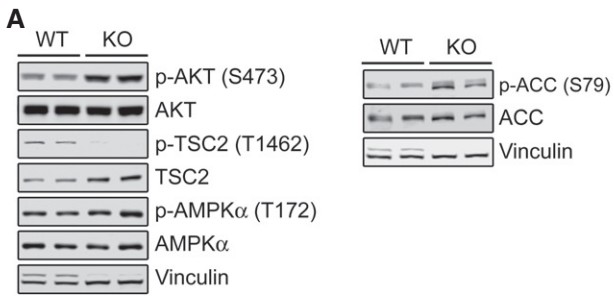

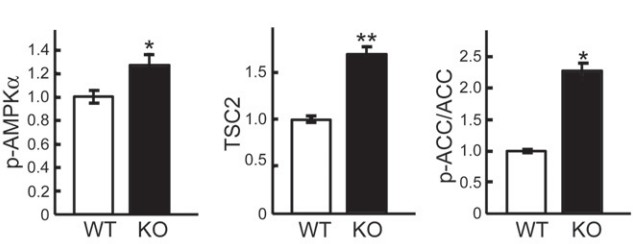

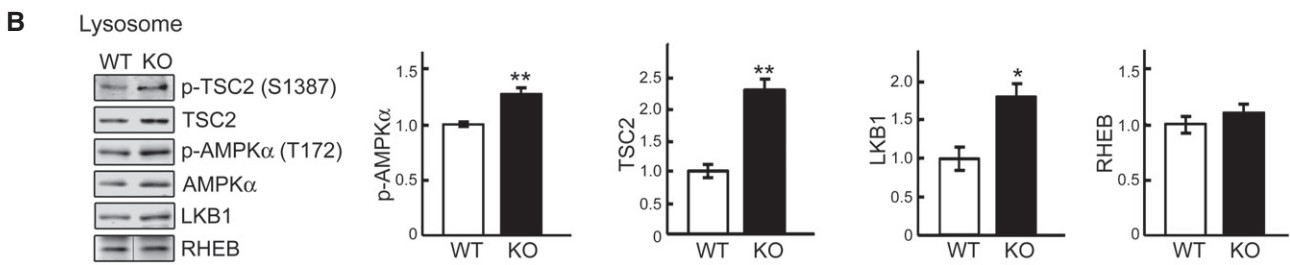

**Figure 3.  Activation of AMPK-TSC2 signaling pathway in KO cells.**

A   Immunoblot analysis of the phosphorylation levels of AKT$^{S473}$, TSC2$^{T1462}$, AMPKα$^{T172}$, and ACC$^{S79}$ in WT and KO cell lysates. Graphs represent mean ± SE. n = 6 for p-AMPKα; n = 3 for TSC2; n = 3 for p-ACC/ACC. *P < 0.05, **P < 0.01, Student's t-test. Vinculin was used as a loading control.
B   WT and KO myotubes were lysed and subjected to fractionation to obtain lysosome-enriched fractions. The isolated fractions were then examined by Western blot showing increased levels of total and p-TSC2$^{S1387}$, total and p-AMPKα$^{T172}$, and total LKB1 in KO cells. Graphs represent mean ± SE. n = 5 for each, p-AMPKα and TSC2; n = 4 for LKB1. *P < 0.05, **P < 0.01, Student's t-test. RHEB was used as a loading control. The blot for RHEB is a composite image; the samples were run on the same gel.

Source data are available online for this figure.

No significant changes in the level of active p-AKT$^{S473}$ were seen in GAA-KO muscle (Fig 4A). Furthermore, the level of phosphorylated PRAS40 (proline-rich AKT substrate of 40 kDa; p-PRAS40$^{T246}$), a downstream target of AKT, was also no different in GAA-KO muscle compared to WT, but the total level of PRAS40 was significantly increased (Fig 4A and B). Because AKT- mediated phosphorylation of PRAS40 is known to relieve the inhibitory effect of PRAS40 on mTORC1 (Sancak *et al*, 2007), an increase in the

amount of hypophosphorylated PRAS40 would lead to the inhibition of mTORC1 activity.

As in cultured KO cells, the phosphorylation levels of p-AMPKα$^{T172}$, the master regulator of cellular energy homeostasis, was increased in GAA-KO compared to those in WT muscle (Fig 4A and B). Furthermore, we have found an elevated ADP/ATP ratio in GAA-KO muscle (Fig 4C), indicating energy deprivation—a condition known to trigger AMPKα activation. The rise in the amount of

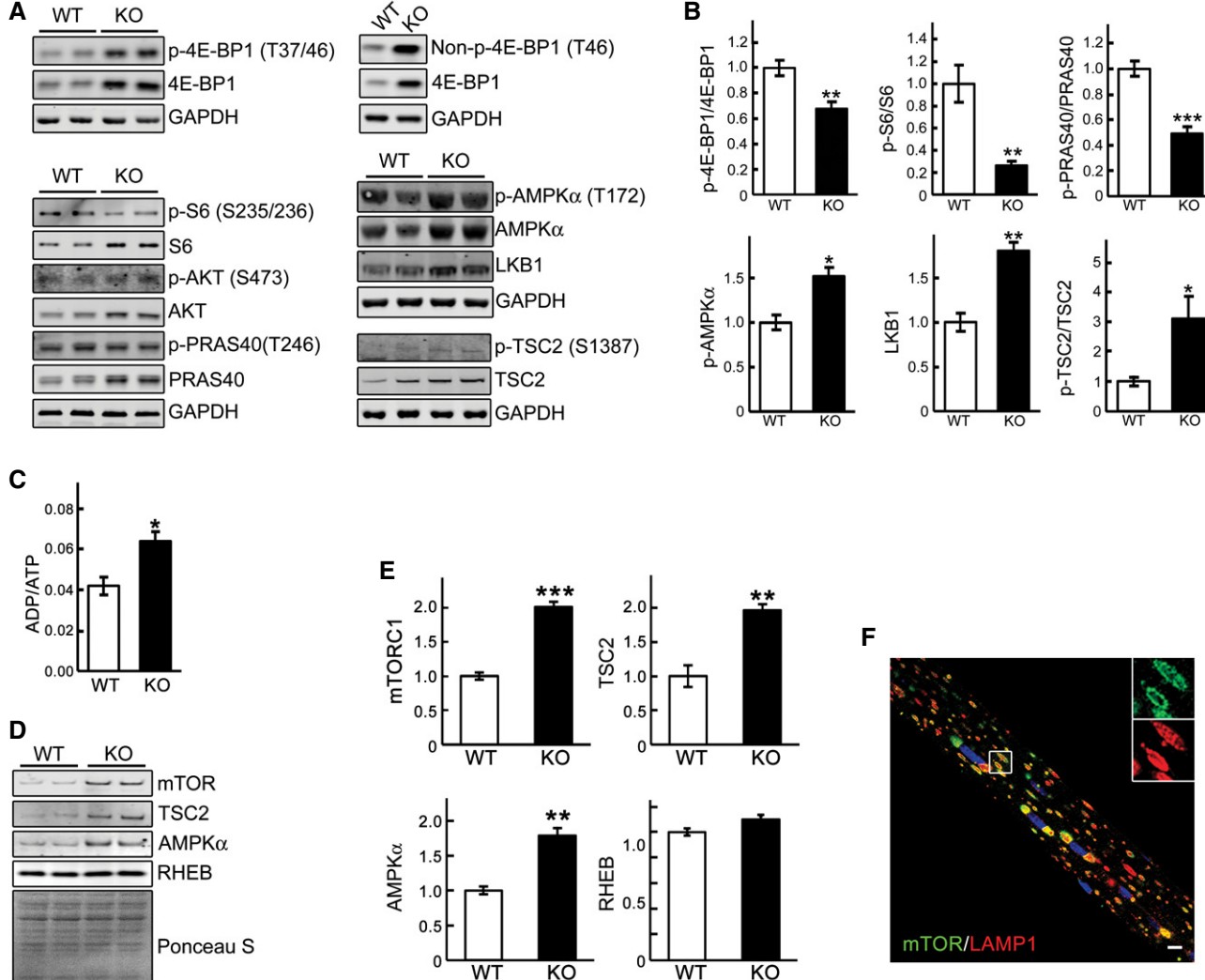

**Figure 4. Disturbance of mTOR signaling *in vivo* in GAA-KO mice.**

Muscle biopsies (white part of gastrocnemius) were obtained from 4- to 6-month-old WT and GAA-KO (KO) mice.

A, B    Western blot analysis of whole muscle lysates from WT and GAA-KO mice with the indicated antibodies. Graphical presentation of the data is shown in (B). Data illustrate the mean ± SE. *n* = 6 for p-4E-BP1/4E-BP1, p-S6/S6, p-PRAS40/PRAS40, and p-TSC2/TSC2; *n* = 5 for p-AMPKα; *n* = 3 for LKB1. *$P < 0.05$, **$P < 0.01$, ***$P < 0.001$, Student's *t*-test. GAPDH was used as a loading control.

C    An increase in ADP/ATP ratio in whole muscle of GAA-KO mice. Data illustrate the mean ± SE. *n* = 3 for WT; *n* = 4 for KO. *$P < 0.05$, Student's *t*-test.

D, E    Muscle tissues derived from WT (*n* = 3) and GAA-KO mice (*n* = 4) were homogenized in lysis buffer and subjected to fractionation to obtain lysosome-enriched fractions. The isolated fractions were then examined by Western blot showing increased levels of total mTOR, TSC2, and AMPKα in GAA-KO. Graphical presentation of the data is shown in (E). Graphs represent mean ± SE. **$P < 0.01$, ***$P < 0.001$, Student's *t*-test. RHEB and Ponceau S staining were used to verify equal protein loading.

F    Immunostaining of a single fiber from a GAA-KO mouse with anti-LAMP1 (red) and anti-mTOR (green) antibodies showing extensive co-localization of the two stains. Scale bar: 10 μm.

Source data are available online for this figure.

active p-AMPKα$^{T172}$ also agrees with both an increase in the level of its upstream activation kinase LKB1 and an increase in the phosphorylation level of its direct downstream target, p-TSC2$^{S1387}$ (Fig 4A and B). Again, as in cultured KO cells, the abundance of AMPKα and TSC2 was increased in lysosome-enriched fraction from GAA-KO muscle (Fig 4D and E). Unexpectedly, the level of mTOR was also increased in the lysosomal fraction, and immunostaining of isolated muscle fibers with mTORC1 and lysosomal marker LAMP1 confirmed a striking co-localization of the two stains (Fig 4D–F). It appears that the activity of mTOR in GAA-KO muscle is reduced despite its excessive accumulation at the lysosome.

Thus, the mechanism of perturbed mTORC1 signaling in GAA-KO muscle and in cultured KO cells is similar in that the suppression of mTORC1 activity is AKT-independent, and that a decrease in the ATP content leads to AMPKα-mediated mTOR inhibition.

### mTOR is locked on the lysosome under nutrient deprivation in KO cells

Recent data have shown that multiple proteins which reside in the cytosol and on the lysosomes are engaged in the recruitment of mTORC1 to the lysosome (activation) and its release from the lysosome (inactivation) (Bar-Peled et al, 2012; Demetriades et al, 2014, 2016; Zhang et al, 2014). We reasoned that lysosomal enlargement and the acidification defect in the diseased muscle cells (Fukuda et al, 2006; Takikita et al, 2009) would affect the interaction of the components of the complex machinery responsible for the proper localization and activation/inactivation of mTORC1. To investigate the relationship between the activity of mTOR and its intracellular localization in Pompe muscle, we again turned to the in vitro model.

As expected, by 2 h of starvation 4E-BP1 and S6 were almost completely dephosphorylated in WT cells; in contrast, the degree of dephosphorylation in the KO was less pronounced, particularly when the cells were treated with medium lacking only amino acids in the presence of dialyzed serum containing growth factors (Fig 5A and B). A weakened mTORC1 response in KO cells is also observed after refeeding subsequent to 2 h of starvation. In WT cells, the phosphorylation of 4E-BP1 after 30 min rebounds to a level that is higher than that at the basal level, whereas in the KO it does not, as shown by the abundance of hypophosphorylated forms in the diseased cells; consistent with this, the levels of non-phosphorylated 4E-BP1 in the KO are much higher than those in the WT at both 15 and 30 min after refeeding (Fig 5C). Of note, the levels of S6K and S6 in the KO were similar to those in WT following refeeding, suggesting a differential effect on 4E-BP1 versus S6K (Fig 5C). This contrary activity of mTORC1 toward its substrates has been reported in other systems (Liu et al, 2004; Choo et al, 2008).

In addition to the control of protein synthesis, mTORC1 regulates autophagy both directly and indirectly by phosphorylating the autophagy-initiating kinase ULK1 (S757) leading to autophagy suppression (Egan et al, 2011; Kim et al, 2011). Rapid dephosphorylation of ULK1 was observed in WT cells after 2 h of starvation leading to a robust induction of autophagy as shown by the increase in LC3-II/LC3 total ratios. As with 4E-BP1 and S6, ULK1 was still phosphorylated in the KO after 2 h of starvation, and autophagy was less robustly induced in KO as shown by a weaker increase in LC3-II/LC3 total ratios (slope 0.96 for WT and 0.34 for KO; Fig 5D). A lesser autophagy response was also observed in the KO cells treated

with a combination of chloroquine and starvation (Appendix Fig S2), consistent with our previous data in WT and KO cells treated with chloroquine or bafilomycin A1 (Lim et al, 2015).

Taken together, the data indicate that KO cells exhibit reduced basal mTOR activity and partial inactivation upon starvation. Next, we looked at the localization of mTOR under these conditions. Immunostaining of myotubes with LAMP1 and mTOR showed co-localization of the two stains in both WT and KO cells, indicating that the recruitment of mTOR to the lysosome is not affected in the diseased cells. However, unlike WT, KO cells failed to release mTOR from the lysosome upon starvation (Figs 6A and EV1A). The immunostaining data are consistent with the Western results showing enhanced lysosomal accumulation of mTOR in fully fed and starved KO myotubes (Fig 6B). The same is true for AMPK and TSC2—both accumulate at the lysosome in excess under basal condition and continue to rise after starvation (Figs 3B and 6C). Notably, there was no difference in the amount of RHEB in the lysosomal fractions from either non-starved or starved KO and WT (Fig 6B), and overexpression of RHEB did not affect the pattern of mTOR localization at the lysosome in these cells (Fig EV2). On the other hand, starvation experiments in KO myoblasts (Fig EV1B), which do not exhibit lysosomal swelling, demonstrated that mTOR did move away from the lysosome following starvation, indicating that a failure to properly modulate mTOR localization and activation/inactivation in the KO cells is associated with the lysosomal enlargement rather than with the enzyme defect itself.

To address the mechanism of mTOR mislocalization following starvation, we treated cells with concanamycin A, a specific inhibitor of the vacuolar proton-translocating V-ATPase that is involved in nutrient sensing (Zoncu et al, 2011). Immunostaining of KO myotubes following starvation in combination with concanamycin A (1 μM) showed mostly cytosolic mTOR staining (Fig 7A). Inactivation of V-ATPase by disassembly of its two major domains, V1 and V0, was also achieved by glucose deprivation as had been reported in non-muscle cells (Sautin et al, 2005; Beyenbach & Wieczorek, 2006; Bond & Forgac, 2008). Immunoprecipitation of V-ATPase from the KO cell lysates using V1-specific antibody followed by Western with V0 antibody resulted in a significantly reduced co-precipitation of V1A/V0C subunits after glucose deprivation, whereas the amount of assembled V-ATPase remained unchanged after starvation (HBSS) (Fig 7B). Disassociation of the V-ATPase subunits, triggered by glucose withdrawal, again led to a release of mTOR from the lysosome in KO cells (Fig 7C), suggesting that activation of V-ATPase may be a mechanism of mTOR retention at the lysosome in the diseased cells. Furthermore, the degree of mTOR inactivation after glucose deprivation was similar in WT and KO as shown by the 4E-BP1 phosphorylation (Fig EV3), indicating that cytosolic localization of mTOR is required for its proper inactivation.

Thus, the link between mTOR activity and its localization appears broken in KO cells: mTOR localizes at the lysosomal surface regardless of its activity or amino acid levels. We hypothesized that in KO cells lysosomal V-ATPase goes into overdrive due to the increased lysosomal pH (Fukuda et al, 2006; Takikita et al, 2009). Therefore, the recruitment of mTOR to the lysosome is not affected in KO cells, but its activity is, due to the constitutive excess of AMPKα and TSC2 (Fig 3B) at the lysosomal surface. mTOR remains

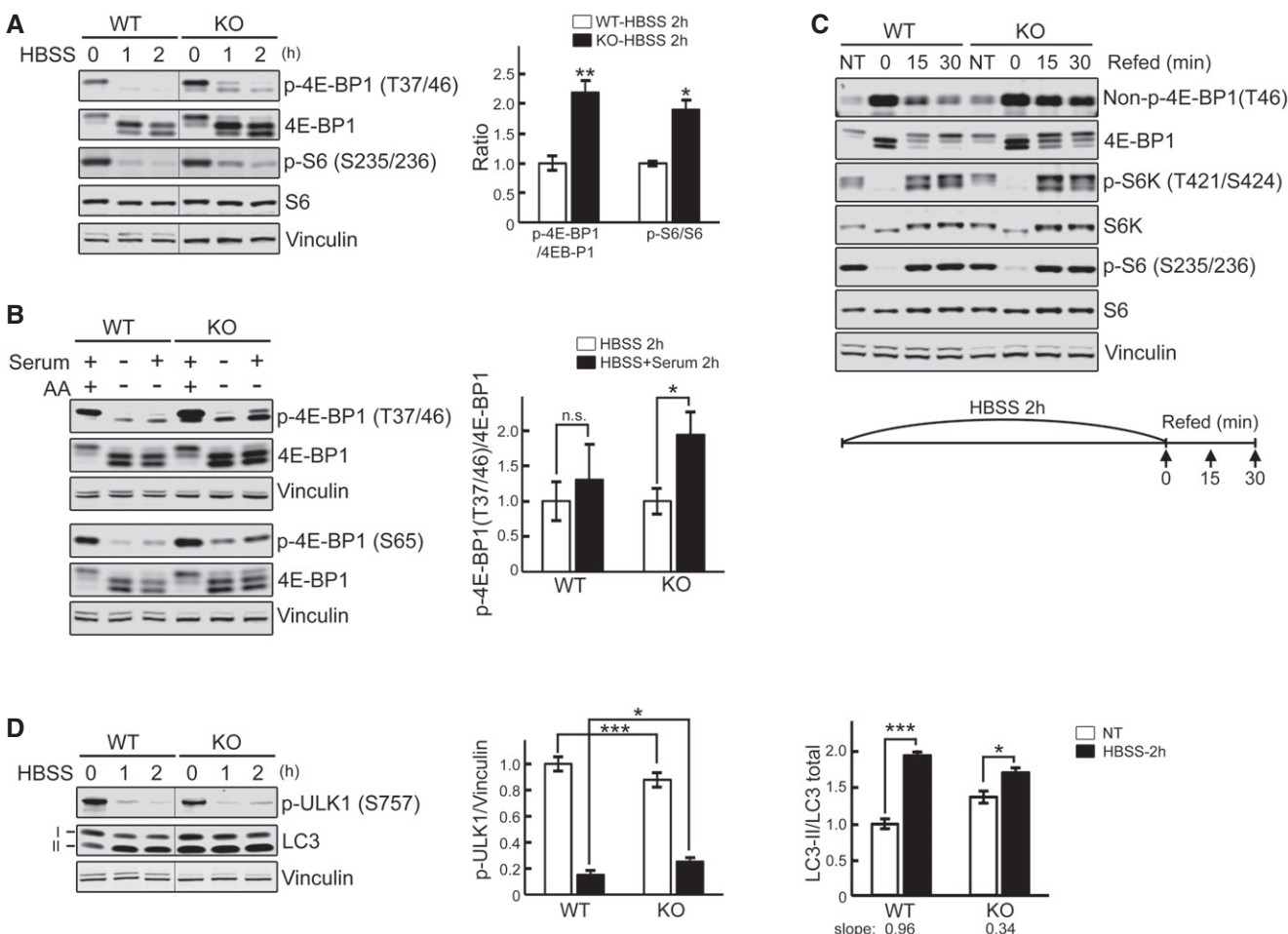

**Figure 5.   KO cells exhibit a diminished response to starvation and inadequate activation after refeeding.**

A   WT and KO myotubes were starved (HBSS) for 0, 1, and 2 h, lysed and subjected to immunoblot analysis with the indicated antibodies. The levels of p-4E-BP1$^{T37/46}$ and p-S6$^{S235/236}$ in KO are higher compared to WT at 1 and 2 h of starvation. Graph shows an increase in p-4E-BP1$^{T37/46}$/total ($n = 5$) and p-S6S$^{235/236}$/total ($n = 3$) ratios in KO compared to WT after 2 h starvation; the data represent mean $\pm$ SE. *$P < 0.05$, **$P < 0.01$, Student's $t$-test. Vinculin was used as a loading control. The blots are composite images; the samples were run on the same gel.

B   WT and KO myotubes were incubated in HBSS with or without dialyzed serum for 2 h, lysed, and subjected to immunoblot analysis with the indicated antibodies. Both p-4E-BP1$^{T37/46}$ and p-4E-BP1$^{S65}$ antibodies were used for the experiments. The degree of 4E-BP1 dephosphorylation after amino acid starvation (AA) is different from that after HBSS in KO (*$P < 0.05$) but not in WT cells (n.s., not significant). Graph represents mean $\pm$ SE; Student's $t$-test. $n = 3$ for each condition.

C   WT and KO myotubes were starved (HBSS) for 2 h, and then refed for 15 and 30 min using differentiation medium as shown schematically. Cell lysates were then subjected to immunoblot analysis with the indicated antibodies. Significant amounts of non-phosphorylated and hypophosphorylated forms of 4E-BP1 are still seen after 15 and 30 min of refeeding in KO cells; the levels of S6K and S6 in the KO were not different from those in controls following refeeding.

D   WT and KO myotubes were starved as in (A). The level of p-ULK1$^{S757}$ in KO is higher than in WT after 2 h starvation. The p-ULK1/vinculin ($n = 4$) and LC3-II/total (LC3-I + LC3-II; $n = 3$) ratios at 2 h after starvation were calculated. Data are mean $\pm$ SE; *$P < 0.05$, ***$P < 0.001$, Student's $t$-test. The blots are composite images; the samples were run on the same gel.

Source data are available online for this figure.

at the lysosome even under nutrient deprivation, again owing to the hyperactivity of the V-ATPase; although the amount of both AMPKα and TSC2 is further increased (Fig 6C), mTOR is not inactivated efficiently because of its excessive accumulation at the lysosome and, perhaps, its proximity to RHEB (Fig 8).

**Restoring mTOR activity in KO cells**

We have used two approaches to reinstate mTOR activity in order to boost protein synthesis in the diseased cells: (i) manipulation of

V-ATPase activity by addressing the lysosomal acidification defect to force proper mTOR localization and (ii) manipulation of TSC2 to relieve its inhibitory effect on RHEB.

To address the acidification defect in the KO cells, we have used polymeric nanoparticles (acNPs; 50 μg/ml; Appendix Fig S3) which traffic to the lysosomal compartment through endocytic pathway and acidify lysosomes upon UV photo-activation (Trudeau *et al*, 2016). To determine whether acNPs affect lysosomal pH in the diseased cells, KO myotubes were exposed to acNPs, UV-treated, and loaded with LysoTracker Red and LysoSensor Green, the dye

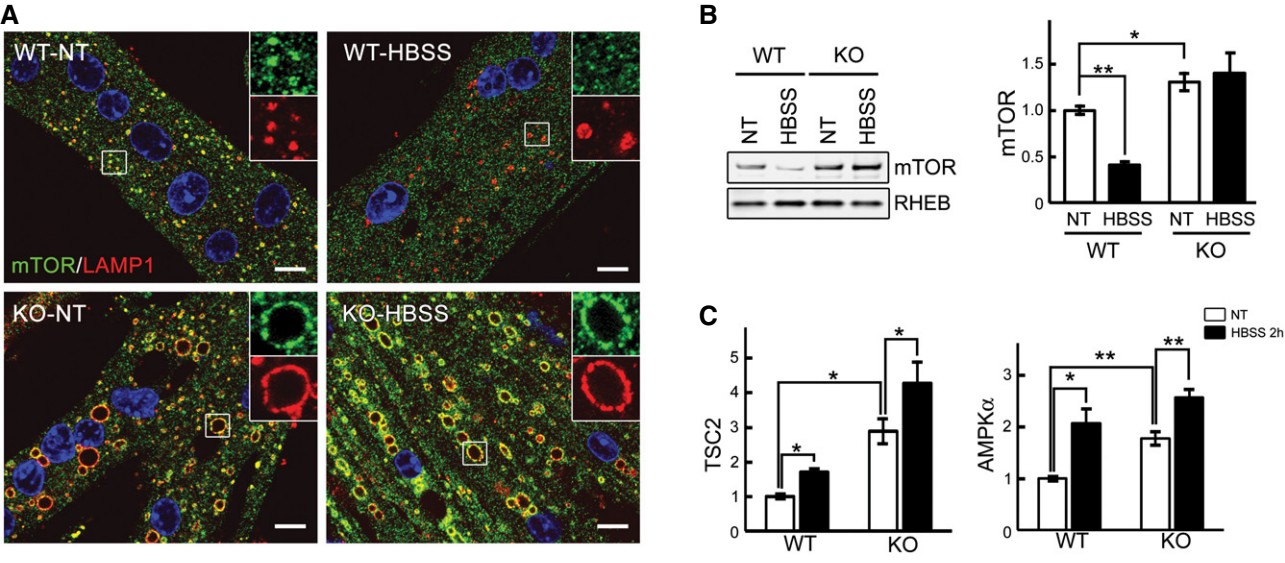

**Figure 6.  mTOR remains at the lysosome in fully fed and starved KO cells.**

A    Immunostaining of WT and KO myotubes with anti-LAMP1 (red) and anti-mTOR (green) antibodies. Lysosomal localization of mTOR is observed in both WT and KO cells grown in differentiation medium (NT, not treated; fed). Mostly cytosolic localization of mTOR (away from lysosomes) is observed in WT cells after 2 h of starvation (HBSS), whereas the two stains still co-localize in KO cells. Scale bar: 10 μm.

B    WT and KO myotubes were lysed and subjected to fractionation to obtain lysosome-enriched fractions. The isolated fractions from fed (NT) and starved (HBSS; 2 h) cells were then examined by Western blot showing increased levels of mTOR in KO cells. RHEB was used as a loading control. Graphs represent mean ± SE; *$P < 0.05$, **$P < 0.01$, Student's *t*-test. $n = 4$ for each condition.

C    WT and KO myotubes were treated as in (B). Graphs show relative amounts of TSC2 and AMPKα in the lysosomal fraction from WT and KO cells in nutrient-rich condition (NT; open bars) and after 2 h of starvation (HBSS; black bars). Data are mean ± SE; *$P < 0.05$, **$P < 0.01$, Student's *t*-test. $n = 3$ for WT; $n = 4$ for KO.

Source data are available online for this figure.

which becomes more fluorescent in acidic environments. The increase in the ratio of green/red fluorescence intensities in acNPs-treated compared to untreated samples indicated that acNPs significantly enhance the population of acidified lysosomes in KO cells (Fig 9A and B). Next, we looked at the effect of acidification on the mTOR localization. KO myotubes were starved for 2 h with or without prior exposure to acNPs followed by fixation and immunostaining with LAMP1 and mTOR. The two stains co-localized after starvation alone (Fig 9C and D; see also Fig 6A), and the degree of co-localization was no different from that in fully fed KO cells. In contrast, pretreatment with acNPs followed by starvation resulted in a decrease in co-localization of the two stains ($P < 0.05$), indicating that at least a portion of mTOR moved away from the lysosome (Fig 9C and D). Consistent with these data, inactivation of mTOR in acNPs-treated KO cells following amino acid starvation was more efficient (Fig 9E).

The second approach included knocking down TSC2 and overexpression of RHEB in KO cells. Infection of myotubes with an adenovirus expressing TSC2 shRNA (Ad-TSC2 shRNA) resulted in efficient shRNA-mediated knockdown of TSC2 (Appendix Fig S4); the effect was evident after 3 days post-infection at $4 \times 10^7$ PFU/ml. Downregulation of TSC2 in KO cells led to increased mTOR signaling, as shown by a significant increase in the levels of p-4E-BP1 and p-S6K (Fig 9F). Similarly, increased mTOR kinase activity was observed in KO myotubes infected with adenovirus expressing RHEB (Ad-RHEB; Fig 9G). Thus, manipulation of TSC2/RHEB may present a promising therapeutic target for Pompe therapy.

### Restoring mTOR activity in GAA-KO mice

An AAV vector expressing TSC2 shRNA (rAAV1-shRNA-TSC1/2) was delivered to 3- to 4-month-old GAA-KO mice ($n = 10$) by a single intramuscular (i.m.) injection in the three sites of the right gastrocnemius muscle with a total dose of $0.75 \times 10^{11}$ vg/muscle. The left gastrocnemius muscle was injected with PBS. Animals were sacrificed at 6–8 weeks after injection. Efficient shRNA-mediated knockdown of TSC2 was observed in all animals, and this inactivation led to an increase in mTOR signaling as indicated by phosphorylation levels of 4E-BP1 (at both S65 and T37/46) and S6. Phosphorylation of AKT and its downstream target, PRAS40, was diminished in the infected compared to the sham-treated muscle (Fig 10A and B), consistent with the negative feedback loop triggered by the activation of mTOR (Bentzinger *et al*, 2008; Castets *et al*, 2013).

Following TSC2 knockdown, the wet weight of gastrocnemius muscle was significantly increased compared to sham-treated contralateral muscle: $183.44 \pm 6.42$ mg and $120.44 \pm 4.07$ mg, respectively ($n = 9$; $P = 7.48 \times 10^{-7}$; Fig 10C). In agreement with our previous data on fiber size of the tibialis anterior and psoas muscles (Raben *et al*, 2008; Takikita *et al*, 2010), GAA-KO gastrocnemius fibers were atrophic [cross-sectional area (CSA) = $736 \pm 12$ μm$^2$] compared to the WT [CSA = $1051 \pm 14$ μm$^2$]. The CSA of the infected fibers was significantly increased compared to the sham-treated contralateral fibers and reached the WT levels ($1191 \pm 25$ μm$^2$; Fig 10D). The reddish color of the infected muscle suggested that mTOR activation caused metabolic changes and fiber

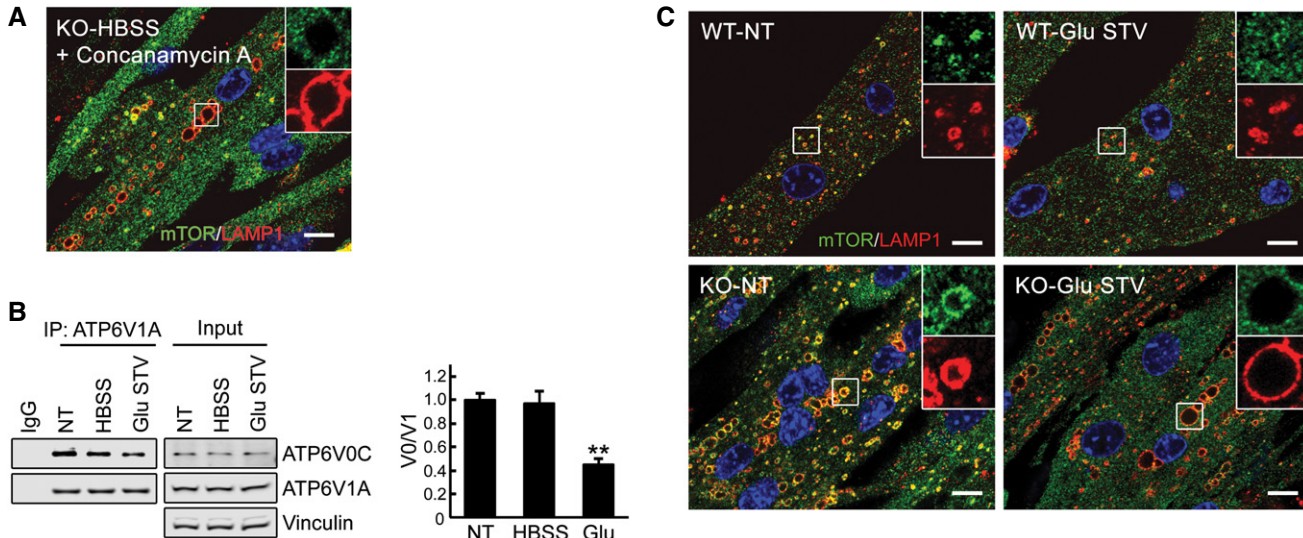

**Figure 7.  Inactivation of V-ATPase in KO cells leads to mTOR release from the lysosome after starvation.**

A    The failure to release mTOR from lysosomes in KO cells is rescued by concanamycin A, an inhibitor of V-ATPase. KO myotubes were starved in the presence of the compound (1 μM), followed by fixation and immunostaining with anti-LAMP1 and anti-mTOR antibodies. The image shows mostly diffused cytosolic localization of the kinase. Scale bar: 10 μm.

B    Glucose starvation inactivates V-ATPase by disassembly of its subunits. Immunoblots of anti-ATP6V1A immunoprecipitates from KO myotubes with ATP6V0C antibodies. A reduced co-precipitation of subunits V1A and V0C is observed after 2 h of glucose starvation (Glu STV), but not after a 2-h HBSS exposure (HBSS). NT, not treated. Representative images of three independent experiments are shown. Graphs represent mean ± SE; **$P < 0.01$, Student's *t*-test.

C    Immunostaining of WT and KO myotubes with anti-LAMP1 (red) and anti-mTOR (green) antibodies. Lysosomal localization of mTOR is observed in both WT and KO cells grown in differentiation medium (NT, not treated; see also Fig 6). Mostly cytosolic localization of mTOR is observed in both WT and KO cells after 2 h of glucose starvation. Scale bar: 10 μm.

Source data are available online for this figure.

type switch. However, no significant shift in the levels of troponin slow, myosin heavy chain fast and slow, and PGC-1α was observed in the infected fibers compared to non-infected control muscle, and no difference in the distribution of the lysosomes was detected (Fig EV4A–C). On the other hand, we did see an increase in the levels of PGC-1β and COX IV, indicating an increase in the mitochondrial mass (Fig EV4A and B).

H&E staining of the infected fibers also revealed a striking absence of "holes", which correspond to the autophagic buildup typically found in the core of muscle fibers from the GAA-KO mice (arrows in Fig 10D). This finding prompted us to analyze LC3-stained fibers isolated from the infected muscle. As shown in Fig 10E, accumulation of the autophagosomal marker was observed in non-infected fibers, but not in the neighboring infected fiber (see also Appendix Fig S5). Consistent with our previous data (Lim *et al*, 2014), autophagic buildup was present in more than 90% of non-treated GAA-KO fibers (*n* = 325; 295 with the buildup), but was only occasionally seen in the infected fibers (*n* = 218; 4 with small areas of the buildup). These data paralleled a decrease in the LC3-II/LC3 total (not shown) and LC3-II/vinculin ratios (Fig 10F). A significant increase in the level of phosphorylated ULK1$^{S757}$ is likely to account for the suppression of autophagy in the infected GAA-KO muscle (Fig 10F).

Thus, genetic inhibition of TSC2 reinstated mTOR activity, reversed the atrophy, and essentially eliminated autophagic buildup in GAA-KO muscle. Coincidentally, arginine has been recently shown to activate mTORC1 by suppressing lysosomal localization of the TSC complex, thus relieving its inhibitory effect on Rheb (Carroll

*et al*, 2016). Considering the site of arginine action, this amino acid seems ideally suited for correction of the defect in the diseased muscle. Therefore, we have attempted to restore mTOR activity in both KO cells and GAA-KO mice by providing excess of L-arginine. Incubation of KO myotubes with L-arginine resulted in a significant increase in mTORC1 activity and a boost in protein synthesis (Figs 11A and B, and EV5). Remarkably, similar effect on the activity of mTOR was achieved *in vivo* in GAA-KO mice (*n* = 5) following the delivery of L-arginine in drinking water (2.5%) over a relatively short (6 weeks) period of treatment (Fig 11C). The attractiveness of arginine treatment is obvious—this amino acid can be taken as a natural dietary supplement.

## Discussion

In general terms, skeletal muscle mass is maintained by a precise dynamic balance between protein synthesis and degradation. Even a small sustained decrease in synthesis or increase in protein breakdown can affect the equilibrium and lead to atrophy (Sandri, 2016). At a molecular level, the reduced rate of protein synthesis is associated with impaired signaling through mTORC1, a major regulator of this anabolic process. This study is the first attempt to systematically analyze mTOR signaling pathway in Pompe disease, an inherited deficiency of lysosomal acid alpha-glucosidase, in which the primary defect—intralysosomal glycogen accumulation—leads to numerous secondary abnormalities including defective autophagy,

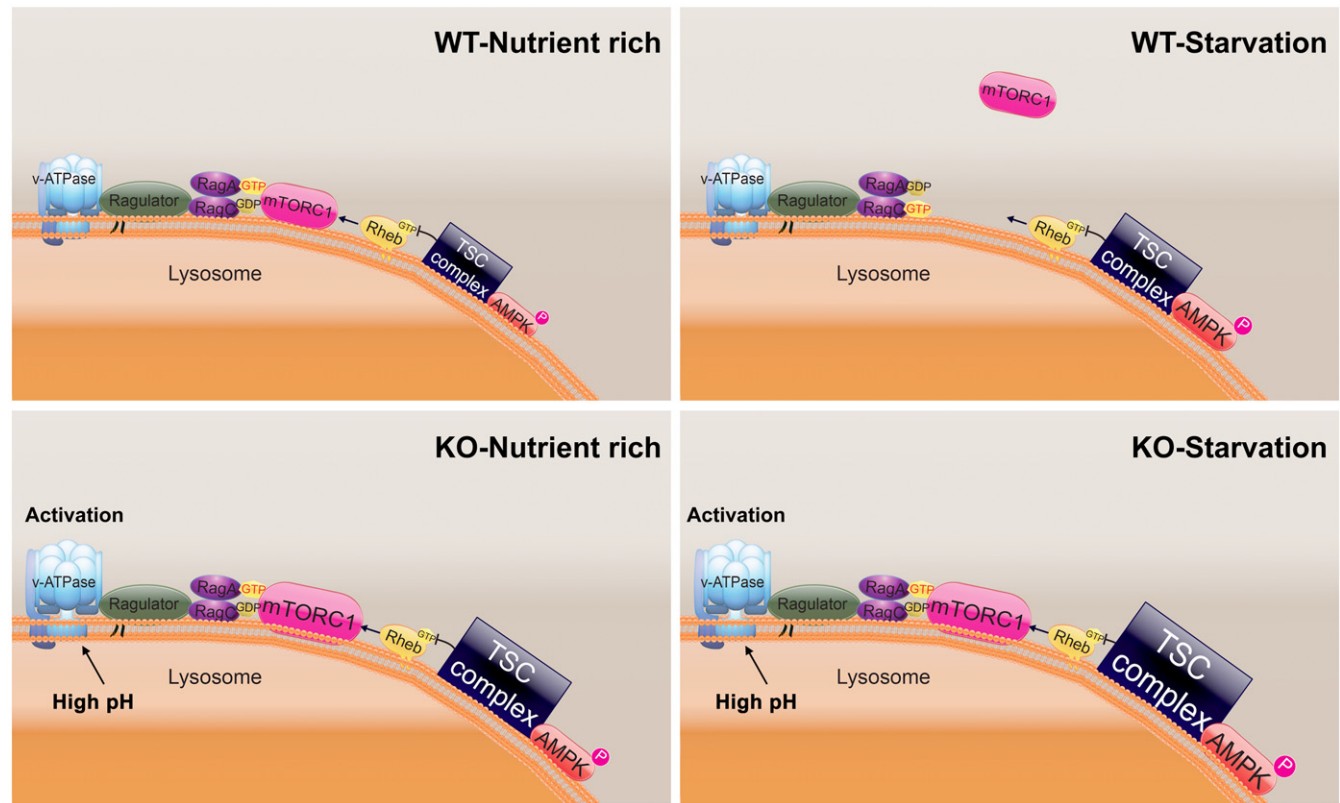

**Figure 8. A putative model of mTOR dysregulation in Pompe disease.**
Activation of V-ATPase in the diseased cells keeps mTOR at the lysosome both under basal (KO-Nutrient rich) and starved (KO-Starvation) conditions. mTOR is not fully active under basal condition due to the inhibitory effect of the excess of TSC2 at the lysosomal surface; the kinase is not fully inactivated in response to starvation due to its proximity to Rheb and despite an additional accumulation of AMPK/TSC2 at the lysosomes.

aberrant mitochondria and calcium homeostasis, and severe muscle wasting (reviewed in Lim *et al* 2014). We have found that the basal activity of mTOR in Pompe muscle cells is reduced; that mTOR is less sensitive to starvation and refeeding; that the relationship between the mTOR activity and its localization is disrupted in the KO cells; and that AMPK-TSC2 pathway is responsible for mTOR inhibition.

Downregulation of mTOR is a shared feature in other muscle disorders. The disturbance of mTOR signaling in Pompe muscle cells is reminiscent of that found in muscle from a mouse model of IBMPFD/ALS (inclusion body myopathy associated with Paget's disease of the bone, frontotemporal dementia, and amyotrophic lateral sclerosis; Ching *et al*, 2013). Reduced mTOR activity is also reported in X-linked myopathy with excessive autophagy (XMEA; Ramachandran *et al*, 2013). An interesting parallel between Pompe disease and XMEA is that both are autophagic myopathies and both exhibit lysosomal acidification defect, but the molecular mechanism of mTOR dysregulation could not be more different. Increased pH in glycogen-laden lysosomes in Pompe cells (Fukuda *et al*, 2006; Takikita *et al*, 2009) leads to V-ATPase activation and mTOR retention at the lysosome. In XMEA, the situation is reversed—the activity of V-ATPase is decreased (due to mutations in VMA21, a protein involved in assembly of V-ATPase subunits), and one can predict that mTOR would remain mostly in the cytosol.

The discovery of a link between the lysosome and mTOR activation/inactivation (reviewed in Laplante & Sabatini, 2012; Demetriades *et al*, 2014) opens a new frontier for investigation into the role of this kinase in the pathogenic cascades in lysosomal storage diseases (LSDs). Reduced basal mTOR activity was shown in several LSDs: in brain and cerebellar cells from $Cln3^{\Delta ex7/8}$ knock-in mice, a model of Juvenile Neuronal Ceroid Lipofuscinosis (JNCL) and in JNCL patient's lymphoblastoid cells (Cao *et al*, 2006); in NPC1- and NPC2-knockdown endothelial cells (Xu *et al*, 2010); in *Drosophila* pupae lacking the TRPML1 homologue (transient receptor potential mucolipin 1), the protein involved in mucolipidosis IV (MLIV; Wong *et al*, 2012); in a human podocyte model of Fabry disease, a deficiency of lysosomal α-galactosidase A (Liebau *et al*, 2013). Impaired mTOR reactivation and defective lysosome reformation were observed in fibroblasts from patients with Scheie syndrome, Fabry disease, and Aspartylglucosaminuria (Yu *et al*, 2010). A more recent study demonstrated downregulation of mTOR and defective lysosomal localization of this kinase after starvation/refeeding in proximal tubular cell lines from $Ctns^{-/-}$ mice, a model of cystinosis, an LSD and the most common cause of renal Fanconi syndrome in children (Gahl *et al*, 2002; Andrzejewska *et al*, 2016).

The information regarding mTOR signaling in Pompe muscle cells is limited to a recent finding of reduced insulin-stimulated mTORC1 activation in two cellular models of Pompe disease—GAA-knockdown C2C12 myoblasts and GAA-deficient human fibroblasts

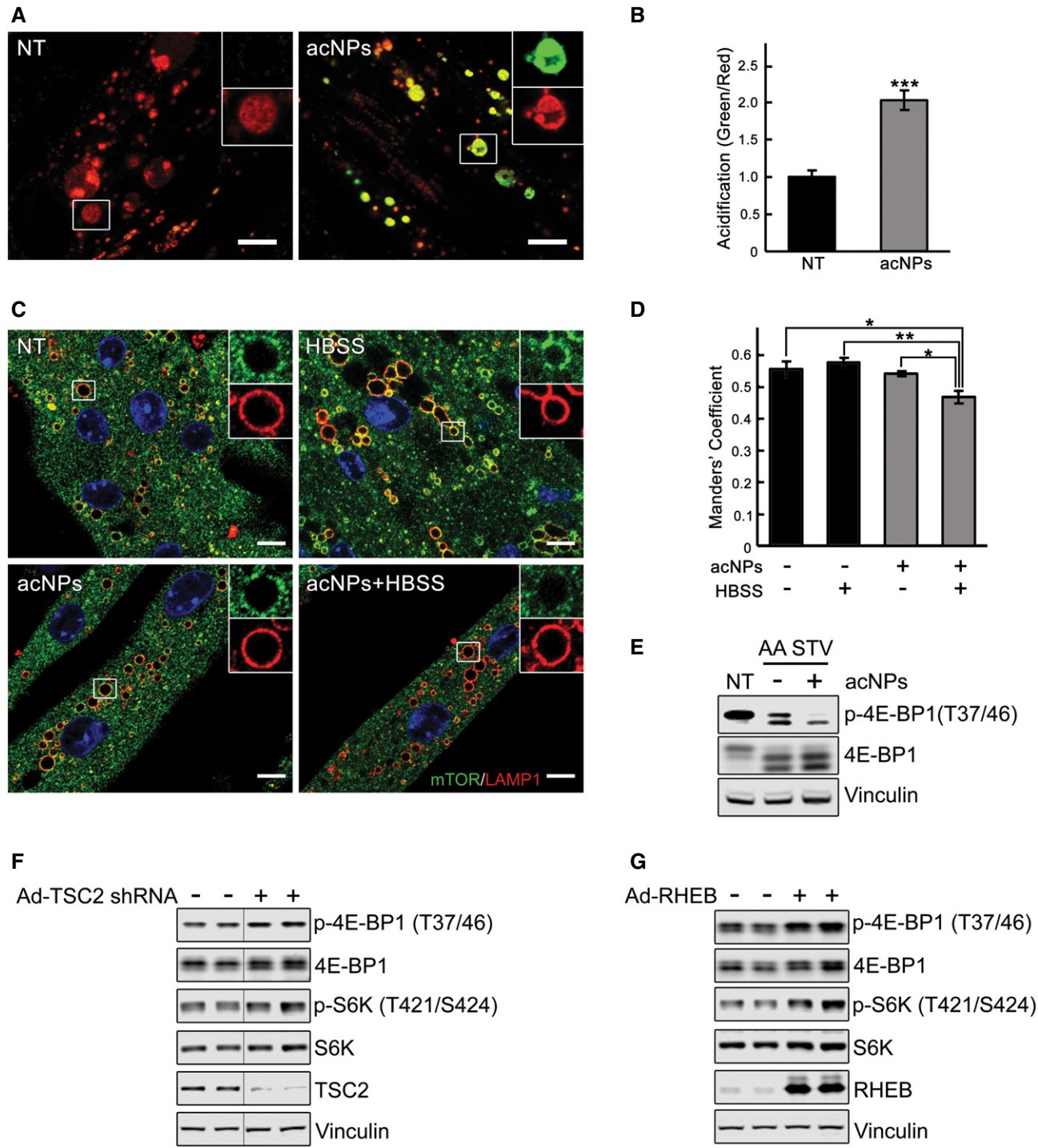

**Figure 9.**

from infantile Pompe patients (Shemesh *et al*, 2014). However, neither basal levels of mTOR activity nor the mechanism of mTOR dysregulation was evaluated in these models, which are quite different from the one used in this study—multinucleated KO myotubes displaying enlarged glycogen-filled lysosomes and defective autophagy—the two major pathologies in the diseased muscle. We have previously demonstrated that both KO myoblasts and fibroblasts from Pompe patients with the most severe infantile form

of the disease show little, if any, difference in the lysosomal size and glycogen content compared to normal cells (Spampanato *et al*, 2013).

We have shown that under basal condition the enlarged lysosomes in KO myotubes can efficiently recruit mTOR, as demonstrated by immunostaining and increased amount of the kinase in the lysosomal fraction from the cell lysates. This suggests that V-ATPase, Ragulator, and the RAGs machinery is more active

**Figure 9.   Rescue of aberrant lysosomal localization and activity of mTOR in KO cells.**

Effect of lysosomal acidification (A–E) and effect of the manipulation of the TSC2/RHEB pathway (F, G).

A, B   KO myotubes were incubated in the presence (acNPs; 50 μg/ml) or in the absence of acNPs (NT) for 18 h and exposed to UV-light for 5 min; following 2 h of incubation in differentiation medium, the cells were loaded with LysoTracker Red and LysoSensor Green (30 min), washed, and visualized by confocal microscopy. Both dyes target lysosomes, but only LysoSensor Green becomes more fluorescent in acidic environments. The acNPs-treated KO cells exhibit a much higher green fluorescence intensity than do untreated cells. Scale bar: 10 μm. The ratios of green/red fluorescence intensities (shown in B) were evaluated using ImageJ software; n = 16 myotubes for KO control; n = 20 myotubes for acNPs-treated KO. Data are mean ± SE; ***$P < 0.001$, Student's t-test.

C, D   KO myotubes were incubated in the presence (acNPs) or in the absence of acNPs (NT) for 18 h, exposed to UV-light for 5 min, and incubated for an additional 1 h in differentiation medium; the cells were then starved for 2 h (HBSS), followed by fixation and immunostaining with LAMP1 (red) and mTOR (green). The degree of co-localization of the two stains (evaluated using ImageJ software; shown in D) is reduced in acNPs-treated cells following starvation. Scale bar: 10 μm. At least five myotubes were analyzed for each condition. Data are mean ± SE; *$P < 0.05$, **$P < 0.01$, Student's t-test.

E   acNPs-treated and non-treated KO myotubes were incubated in HBSS with dialyzed serum for 2 h, lysed and subjected to immunoblot analysis with the indicated antibodies; the degree of dephosphorylation of 4E-BP1 is more prominent in acNPs-treated cells. Vinculin was used as a loading control.

F, G   Immunoblot analysis showing the effects of TSC2 inhibition (F) and RHEB overexpression (G) on mTOR activity in KO myotubes. Increased phosphorylation of the two major downstream mTOR targets, 4E-BP1 and S6K, is observed following the treatments. Vinculin was used as a loading control. Blots in (F) are composite images; the samples were run on the same gel.

Source data are available online for this figure.

compared to controls. However, the activity of mTOR is less than in controls because of the inhibitory effect of TSC2. Activation of TSC2 in the KO fits with the activation of AMPK, and both proteins are found in excess at the lysosome in the KO. The other part of the process—mTOR transport back to the cytosol and inactivation—is profoundly affected in the KO cells. The enlarged lysosomes fail to release mTOR upon starvation. The retention and even further accumulation of mTOR at the lysosome under starvation is a striking feature of the KO cells, again suggesting an increased activity of V-ATPase. Under starvation conditions, mTOR is expected to dissociate from the lysosome and become inactive; instead, the kinase still remains in partially active state in the KO cells. The inability to efficiently shut down protein synthesis under starvation may have dire consequences for the cell. The condition may be exacerbated in the KO, because unlike in control cells, autophagy is only weakly upregulated; however, it is important to emphasize that the increased basal autophagy in KO cells may contribute to a lesser response.

Understanding the mechanism of the disturbed mTOR signaling in Pompe muscle cells opens the possibility for novel treatment strategies, which are much needed as the presently available therapy, replacement of the missing enzyme, is unable to reverse muscle pathology (reviewed in Lim et al, 2014). Re-acidification of lysosomes is an attractive approach for two reasons: (i) it would allow mTOR to move freely to and from the lysosome, thus facilitating its proper regulation, and (ii) it would enhance the effect of the therapeutic enzyme (recombinant human GAA; alglucosidase alfa, Genzyme Corp, a Sanofi Company), which functions optimally at acidic pH. Acidic nanoparticles have been recently shown to restore lysosomal function in cellular models of Parkinson's disease and in XMEA-mutant fibroblasts (Bourdenx et al, 2016).

Restoration of mTOR activity through amino acid supplementation, especially leucine, has received much attention as potential therapy for muscle wasting. In general, the importance of maintaining an adequate dietary intake of leucine is well-established, and of all the amino acids, leucine, indeed, is the most effective positive regulator of protein synthesis via mTORC1 pathway in culture systems (Kimball & Jefferson, 2006; Norton & Layman, 2006). The mixture of amino acids lacking leucine induced lysosomal localization of mTOR, but failed to fully activate the kinase (Averous et al, 2014). However, in vivo studies produced conflicting reports (Dodd

& Tee, 2012). Additional leucine uptake did not prevent the decrease in lean mass in aging rats (Vianna et al, 2012), and long-term leucine supplementation showed no beneficial effect on muscle mass and function in placebo-controlled studies in humans (reviewed in Ham et al, 2014). In addition, the dose-dependent increase in liver and muscle glycogen was reported in trained rats receiving chronic supplementation of branched-chain amino acids (de Araujo et al, 2006).

Irrespective of these somewhat conflicting conclusions, emerging evidence indicates that the signaling pathways that connect individual amino acids to mTORC1 activation at the lysosome are different (Jewell et al, 2015). Leucine stimulates Rag GTPase-mediated translocation of mTORC1 to lysosomal membranes and this mechanism of mTORC1 activation is TSC-independent (Sancak et al, 2010). Recent data demonstrate that leucine enters the lysosome through the leucine transporter, which is recruited to the lysosomal membrane by the lysosomal protein LAPTM4b (Milkereit et al, 2015). Leucine supplementation has been recently proposed as a therapeutic approach for Pompe disease (Shemesh et al, 2014). Our data demonstrate the leucine-mediated process of mTOR recruitment to the lysosome is already activated in the diseased Pompe muscle cells; therefore, pushing it further does not seem to make much sense, and may even be harmful in the long run.

It appears that AMPK-TSC2 pathway is a much better therapeutic target in Pompe disease since the increased activity of these proteins and their excess at the lysosome are responsible for mTOR inhibition. Indeed, we have shown, as a proof of principle, that knocking down TSC2 or overexpressing RHEB enhances mTOR activity in KO cells. Similar effect of Rheb overexpression was observed in VCP-IBM muscle; activation of mTOR in this model led to amelioration of muscle atrophy (Ching et al, 2013). Increased myofiber size was also observed in murine muscle following transient expression of Rheb (Goodman et al, 2010). Restoring mTOR activity by overexpression of constitutively active Rheb showed beneficial effect not only in muscle, but also in neurological disorders, such as Huntington's disease (Lee et al, 2015).

A cautionary note on the excessive mTOR activation, at least in skeletal muscle, comes from the study, in which sustained activation of mTORC1 by genetic TSC deletion caused late-onset myopathy (Bentzinger et al, 2013). It is, however, important to emphasize that the underlying goal in Pompe disease is not to achieve constant

**Figure 10.    AAV-mediated TSC2 inhibition in muscle of the GAA-KO mice activates mTOR, increases muscle mass, and eliminates autophagic buildup.**

A, B    Western blot analysis of whole muscle lysates from sham-treated (NT) left gastrocnemius and AAV1-shRNA-TSC2 (AAV-shTSC2)-infected right gastrocnemius muscle from GAA-KO mice (*n* = 7) with the indicated antibodies. Graphical presentation of the data is shown in (B). Data illustrate the mean ± SE; *\*P* < 0.05, *\*\*\*P* < 0.001, Student's *t*-test. All blots (except for S6; *n* = 3) are representative of seven independent experiments.

C    Images show an increase in muscle mass in AAV-shTSC2-infected (bottom) muscle compared to sham-treated (top) muscle. Intramuscular injections were performed in 3- to 4-month-old GAA-KO mice; the animals were sacrificed 7–8 weeks after injections (*n* = 9). Data are mean ± SE; *\*\*\*P* < 0.001, Student's *t*-test.

D    H&E-stained sections of WT, sham-infected (KO: NT), and infected (KO: AAV-shTSC2) GAA-KO gastrocnemius muscle reveal an increase in cross-sectional area (CSA) in the infected fibers, and a paucity of large vacuoles (arrows) typically present in the GAA-KO muscle fibers. Scale bar: 20 μm. Histogram showing the distribution of CSAs of WT, sham-treated, and infected GAA-KO muscle; note that CSAs of a subset of the infected fibers (~15%) are larger than those in the WT. *n* = 240 for each of the three conditions. Data are mean ± SE; *\*\*\*P* < 0.001, Student's *t*-test.

E    Immunostaining of isolated muscle fibers with autophagosomal marker LC3 (red); the fibers were derived from AAV1-shTSC2-infected gastrocnemius muscle of a GAA-KO mouse. Of the four adjacent fibers, only one was infected (left panel; green); autophagic accumulations are seen in non-infected, but not in the infected fiber (right panel). Note that the infected fiber is bigger in size. Scale bar: 20 μm. Data are mean ± SE; *\*\*\*P* < 0.001, Student's *t*-test. *n* = 227 of non-infected fibers; *n* = 139 of infected fibers.

F    Western blot analysis of whole muscle lysates from sham-treated (NT) left gastrocnemius and AAV-shTSC2-infected right gastrocnemius muscle from GAA-KO mice (*n* = 7) with the indicated antibodies. An increase in p-ULK1$^{S757}$/total ratio (mTOR-mediated phosphorylation) and a concomitant decrease in LC3-II/vinculin ratio are seen in AAV-shTSC2-infected muscle. Data are mean ± SE; *\*\*P* < 0.01, *\*\*\*P* < 0.001, Student's *t*-test.

Source data are available online for this figure.

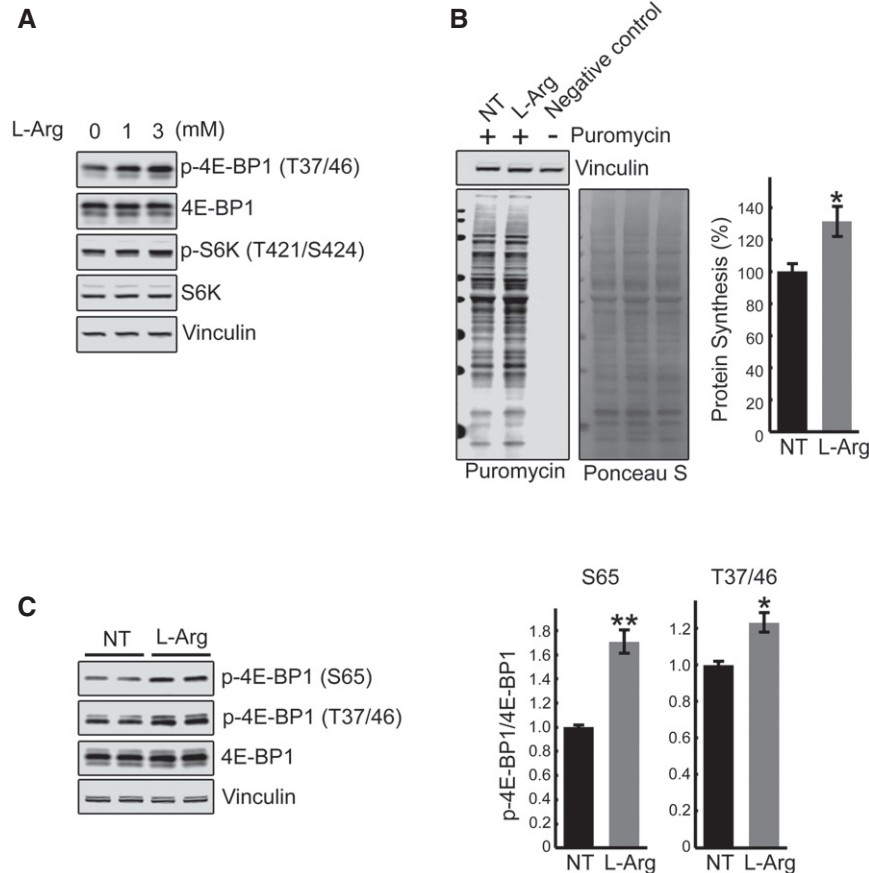

**Figure 11. ʟ-arginine treatment activates mTOR both *in vitro* and *in vivo*.**

A Immunoblot analysis showing the effects of arginine treatment on mTOR activity in KO myotubes. Increased phosphorylation of the two major downstream mTOR targets, 4E-BP1 and S6K, is observed following the treatments.

B SUnSET analysis was used to evaluate the incorporation of puromycin into nascent polypeptides. KO myotubes were treated with both ʟ-arginine (3 mM) and puromycin (1 μM) for 15 min. Western blot with anti-vinculin antibody and Ponceau S staining were used as loading controls. Total intensity of puromycin-labeled polypeptides was quantified. Data are mean ± SE of three independent experiments. *$P < 0.05$, Student's *t*-test.

C GAA-KO mice were treated with ʟ-arginine (2.5% in drinking water) for 6 weeks. Western blot analysis of whole muscle lysates showing an increase in p-4E-BP1$^{S65}$/total and p-4E-BP1$^{T37/46}$/total ratios following dietary arginine supplementation. Graphs represent mean ± SE; *$P < 0.05$, **$P < 0.01$, Student's *t*-test. $n = 3$ for the NT; $n = 5$ for ʟ-Arg.

Source data are available online for this figure.

hyperactivation of this kinase, but rather to restore its diminished activity. Furthermore, the late-onset myopathy described in muscle-specific TSC knockout mice was linked to the mTOR-dependent inhibition of autophagy through ULK1, and the pathology was reminiscent of that in autophagy-deficient muscle (Bentzinger *et al*, 2013; Castets & Ruegg, 2013). We, too, observed mTOR-dependent inhibition of ULK1 in GAA-KO muscle following TSC knockdown, but in the setting of Pompe disease, this suppression of autophagy turned out to be beneficial as shown by the removal of the autophagic buildup. We have previously shown that the poor muscle response to ERT in Pompe disease was associated with the presence of this buildup and that its elimination rendered muscle amenable to ERT (Raben *et al*, 2010).

The aspiration is to identify a compound which would enhance mTOR activity in Pompe muscle cells by interfering with TSC2-Rheb signaling. Recent data indicate that arginine does exactly that; it activates mTORC1 by promoting dissociation of TSC from lysosome,

thus relieving its inhibition of Rheb (Carroll *et al*, 2016). Remarkably, a short exposure of both KO cells and GAA-KO mice to arginine restored mTOR activity.

Thus, we have identified promising therapeutic targets to reverse aberrant mTOR signaling in Pompe disease. Modulation of TSC2-Rheb pathway by amino acid arginine is particularly attractive since this approach can be added "risk-free" to the current therapy in the clinical setting to combat muscle wasting in Pompe disease. Similar approach may well prove beneficial to a whole range of muscle wasting conditions and lysosomal storage diseases.

## Materials and Methods

### Reagents

The information is provided in the Appendix file.

## Animal model, muscle tissue processing and analysis, and treatment

Acid alpha-glucosidase knockout mice [GAA-KO; a mouse model of Pompe disease (Raben *et al*, 1998)] were used for the experiments. Gender-matched WT and GAA-KO mice were used at the age of 4–6 months (unless indicated otherwise). White part of gastrocnemius muscle was used for the experiments.

Lysosomal fraction from muscle homogenates was prepared as described (Christensen *et al*, 2003) with some modifications. Briefly, muscle was homogenized in buffer containing 10 mM HEPES (pH 7.0), 0.25 M sucrose, 1 mM EDTA, and protease/phosphatase inhibitors (Cell Signaling, 5872S) using a Dounce homogenizer. A low-speed (nuclear) fraction was pelleted at $800 \times g$ for 10 min at 4°C; the procedure was repeated 3 times. The supernatant was pooled and centrifuged at $100,000 \times g$ for 1 h in a 70Ti rotor (Beckman, Palo Alto, CA). The high-speed pellet was re-suspended in 1 ml of homogenization buffer, mixed with 7 ml of 16% (vol/vol) Percoll (Sigma; P1644), and 250 μl of Percoll was then added beneath the mix using a syringe. Following centrifugation at $60,000 \times g$ for 30 min, 1.5 ml fraction at the bottom of the tube was collected, diluted 1:5 in homogenization buffer, and centrifuged again at $15,000 \times g$ for 10 min at 4°C; the pellet constitutes a lysosome-enriched fraction, which was analyzed by Western blotting. ATP extraction from muscle tissues was performed using a phenol-based method as described (Chida *et al*, 2012). ADP/ATP ratio in the extracts was then measured using ADP/ATP Ratio Assay Kit (Sigma, MAK135) according to the manufacturer's instructions. LAMP1/mTOR immunostaining of single muscle fibers was performed as previously described (Raben *et al*, 2009).

Ten 3- to 4-month-old $GAA^{-/-}$ mice were used for intramuscular injection of rAAV1-shRNA-TSC1/2 with a total dose of $0.75 \times 10^{11}$ vg/muscle. The vector preparation was injected into three sites of the right gastrocnemius (three injections of $0.25 \times 10^{11}$ vg in 25 μl each) muscle using a Hamilton syringe. Equal volumes of PBS were injected into the contralateral muscle. Mice were sacrificed 7–8 weeks after injections. For fiber size measurements, gastrocnemius muscle was fixed in 4% paraformaldehyde, cross-sectioned, and stained with H&E. Muscle fiber size was measured in WT, GAA-KO infected, and GAA-KO sham-infected muscle fibers; for each condition, 240 fibers were counted. The quantifications were done with ImageJ. Two mice of each genotype were used for the experiments. 2.5-month-old GAA-KO mice ($n = 5$) were receiving L-arginine supplementation (2.5% in drinking water) over the course of 6 weeks.

Animal care and experiments were conducted in accordance with the National Institutes of Health Guide for the Care and Use of Laboratory Animals.

## Muscle cell cultures, treatment, processing and analysis

The generation of immortalized WT and KO (clones 6 and RF1) mouse muscle cells has been previously described (Spampanato *et al*, 2013). The cells were grown on Matrigel (Corning; 354234)-coated dishes at 33°C in an atmosphere of 5% $CO_2$ in proliferation medium [20% fetal bovine serum, 10% horse serum, 1% chick embryo extract, recombinant IFN-γ (100 U/ml; Life Technologies), 1× penicillin/streptomycin/L-glutamine in high-glucose (4.5 g/l)

DMEM]. When myoblasts became nearly confluent, the medium was changed to differentiation medium [DM; DMEM containing 2% horse serum, 0.5% chick embryo extract, recombinant human insulin (10 μg/ml, Life Technologies, 12585–014), 1× penicillin/streptomycin/L-glutamine], and the cells were moved to 37°C in an atmosphere of 5% $CO_2$. Myotubes began to form within 3–4 days.

For starvation experiments, myotubes (on days 7–11 in DM) were washed and incubated for 2 h at 37°C in either Hanks' balanced salt solution (HBSS; amino acids and growth factors starvation) or glucose-free DMEM (Invitrogen, 14430). For refeeding experiments, myotubes were incubated for 2 h in HBSS, followed by incubation in DM for 15 and 30 min. For amino acid starvation, myotubes were incubated for 2 h in HBSS in the presence of 2% dialyzed serum (Life Technologies, 26400-036) containing growth factors but lacking amino acids.

For drug treatment experiments, myotubes were incubated at 37°C for 2 h in HBSS containing 1 μM concanamycin A (Santa Cruz, sc-202111); alternatively, myotubes were incubated at 37°C for 15 min in DM containing L-arginine (Sigma; A4474) at a concentration of 1, 3, and 5 mM. For the evaluation of autophagy, myotubes were treated with chloroquine (6 h; 50 μM) in combination with HBSS (2 h); the cells were fixed in acetone–methanol mix (equal volumes) at −20°C for 15 min, washed twice in PBS, and immunostained with LAMP1 and LC3. For lysosomal acidification experiments, KO myotubes (on day 8 in DM), grown in Matrigel-coated 8-well chambers (ibidi; 80826), were treated with acidic nanoparticles (acNPs; 50 μg/ml in DM) for 18 h. UV photoactivation (UV lamp model: ENF-260C, Spectronics Corporation; the lamp has long-wave UV at 365 nm, 115 V, and 60 Hz) was performed for 5 min at room temperature with plate lids removed. The cells were then allowed to recover for 1 h in DM, starved for 2 h, fixed in 2% paraformaldehyde (Electron Microscopy Sciences, Hatfield, PA) for 15 min at room temperature, washed twice in PBS, and immunostained with LAMP1 and mTOR. The extent of LAMP1/mTOR co-localization was evaluated with ImageJ software using the Manders' algorithm. Alternatively, following exposure to acNPs and UV-activation, KO cells were allowed to recover for 2 h in DM, loaded with 50 nM LysoTracker Red (DND-99) and 1 μM LysoSensor Green (DND-189) for 30 min, washed, and analyzed by confocal microscopy; UV-irradiated/acNPs-untreated KO cells were used as controls. LysoSensor Green (DND-189) accumulates in acidic organelles and exhibits green fluorescence.

Myotubes (on days 4–5 in DM) were infected with adenovirus expressing either mTSC2 shRNA or mRHEB for 3 days. The cells were then washed, homogenized in RIPA buffer [PBS containing 1% NP-40, 0.5% sodium deoxycholate, 0.1% SDS, and a protease/phosphatase inhibitor cocktail], centrifuged for 10 min at $18,000 \times g$ at 4°C, and the supernatant was used for Western blots.

For isolation of lysosomal fraction, cells were grown in Matrigel-coated 6-well plates or 100-mm dishes, lysed, and processed as described above. For immunostaining, fixed myotubes were permeabilized in 0.2% Triton X-100 (Sigma-Aldrich, St. Louis, MO), and staining was done using M.O.M. kit (Vector Laboratories, Burlingame, CA) as previously described (Raben *et al*, 2009). The cell nuclei were stained with 2 μg/ml Hoechst 33342 (Life Technologies) in PBS for 10 min. After staining, the cells were imaged on a Carl Zeiss LSM 780 confocal microscope with a 40× or 63× oil immersion objective.

## Immunoprecipitation and Western blot analysis

Myotubes were homogenized in RIPA buffer (see above). Dynabeads® Protein A were prepared according to the manufacturer's instructions (Invitrogen; 10001D). For pre-clearing, equal amounts of protein were incubated with rabbit IgG-conjugated Dynabeads® for 4 h at 4°C. The supernatant was incubated with primary antibody for overnight at 4°C, followed by incubation with Dynabeads® for 4 h at 4°C; the supernatant was discarded, and the beads were washed 3 times with RIPA buffer; the immunoprecipitant was eluted with SDS sample buffer, followed by Western analysis.

For Western blot analysis, cell lysates or whole muscle tissues were homogenized in RIPA buffer. Samples were centrifuged for 10 min at $18,000 \times g$ at 4°C. Protein concentrations of the supernatants of the total lysates were measured using the Bio-Rad Protein Assay (Bio-Rad Laboratories, Inc.). Equal amounts of protein were run on SDS–PAGE gels (Invitrogen, Carlsbad, CA) followed by electro-transfer onto nitrocellulose membranes (Invitrogen, Carlsbad, CA). Membranes were blocked in 1:1 PBS and Odyssey Blocking Buffer (LI-COR Biosciences, Lincoln, NE), incubated with primary antibodies overnight at 4°C, washed, incubated with secondary antibodies and washed again. Blots were scanned on an infrared imager (LI-COR Biosciences).

## Measuring the rate of protein synthesis in muscle cells

Protein synthesis in WT and KO myotubes was evaluated using surface sensing of translation (SUnSET) method as described (Goodman *et al*, 2011). Briefly, the cells were incubated with serum-free DMEM for 90 min followed by incubation with 1 μM puromycin (an analogue of tyrosyl-tRNA; Invitrogen; A11138-03), for 30 min. The amount of puromycin incorporated into nascent peptides was then evaluated by Western blot using antibody to puromycin.

## Statistical analysis

Statistical significance was determined by two-tailed Student's *t*-test; error bars represent SE. *$P < 0.05$ was considered statistically significant. ** indicate *P*-values $< 0.01$; *** indicate *P*-values $< 0.001$. The list of actual *P*-values for each test is provided in Table EV1.

**Expanded View** for this article is available online.

## Acknowledgements
We would like to thank Kristien Zaal and Evelyn Ralston (Light Imaging Section, Office of Science and Technology, NIAMS, NIH) for their help with the imaging. We are grateful to Dr. Markus A. Rüegg (Biozentrum, University of Basel) for the generous gift of shRNA-TSC1/2 plasmid. This research was supported in part by the Intramural Research Program of the National Institute of Arthritis and Musculoskeletal and Skin diseases of the National Institutes of Health. Dr. Lim and Dr. Li are supported in part by a CRADA between NIH and Genzyme Corporation and from the Acid Maltase Deficiency Association.

## Author contributions
J-AL performed tissue culture/whole muscle experiments, analyzed and interpreted the data, and participated in preparation of the manuscript; LL performed *in vivo* experiments, analyzed the data; OSS and KMT contributed new reagents and analytical tools, interpreted and analyzed data; RP analyzed and interpreted the data and participated in writing of the manuscript; NR designed, interpreted, and analyzed data, and wrote the manuscript.

## Conflict of interest
The authors declare that they have no conflict of interest.

## The paper explained

### Problem
Pompe disease, a deficiency of glycogen-degrading lysosomal enzyme acid alpha-glucosidase, is an inherited neuromuscular disorder that causes progressive muscle loss. Mammalian target of rapamycin (mTOR), a master regulator of cellular growth and metabolism, is directly involved in the control of muscle mass. The relevance of mTOR signaling to the pathogenesis of this lysosomal storage disorder is further emphasized by recent well-documented data showing a link between mTOR activity and lysosomal function. This study details the first systematic analysis of mTOR pathway in Pompe muscle cells by evaluating mTOR activity, localization, regulation in response to nutrients, and its role in the control of protein synthesis and autophagy.

### Results
Experiments in muscle cell culture and an animal model of Pompe disease demonstrate that the diseased muscle cells exhibit reduced mTOR activity, defective dephosphorylation of its key substrates, and inability to displace mTOR from lysosome upon nutrient deprivation. Excessive lysosomal accumulation of TSC2 is responsible, at least in part, for mTOR signaling defect. Knockdown of TSC2 in Pompe mice led to mTOR reactivation, reversal of muscle atrophy, and a striking removal of the autophagic buildup that is typically present in the affected fibers along with glycogen-filled enlarged lysosomes. Activation of mTOR signaling was also achieved by L-arginine supplementation.

### Impact
Experimental data and clinical experience indicate that the currently available enzyme replacement therapy for Pompe disease has only partial efficacy in skeletal muscle. The limitations dictate the need for new adjunctive treatments. Our study uncovers novel therapeutic targets and provides evidence that activation of mTOR signaling in the diseased muscle can reduce autophagic defect and prevent a decline in muscle mass. Reinstatement of mTOR activity by arginine may prove an effective strategy for Pompe disease as well as for a large group of neuromuscular and lysosomal disorders.

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
