## [Review Process File · EMBO Molecular Medicine]

Modulation of mTOR signaling as a strategy for the treatment of Pompe disease

Jeong-A Lim, Lishu Li, Orian Shirihai, Kyle M. Trudeau, Rosa Puertollano, and Nina Raben

*Corresponding authors: Nina Raben, Lishu Li NIAMS, National Institutes of Health
Rosa Puertollano, National Heart, Lung, and Blood Institute, National Institutes of Health*

Review timeline:

Submission date:	26 April 2016
Editorial Decision:	29 April 2016
Resubmission:	23 August 2016
Editorial Decision:	26 September 2016
Revision received:	03 November 2016
Editorial Decision:	06 December 2016
Revision received:	12 December 2016
Editorial Decision:	14 December 2016
Revision received:	15 December 2016
Accepted:	16 December 2016

Transaction Report:

Editor: Roberto Buccione

1st Editorial Decision

29 April 2016

Thank you for the submission of your manuscript "Dissecting the mechanisms of mTOR dysregulation in Pompe disease".

I have now had the opportunity to carefully read your paper and the related literature and I have also discussed it with my colleagues. I am afraid that we concluded that the manuscript is not well suited for publication in EMBO Molecular Medicine and have therefore decided not to proceed with peer review.

Although we acknowledge the high quality of your experimentation and the potential interest of your findings, we find your work to be better suited to a specialist readership. In fact, while we appreciate the dissection of the consequences of GAA KO on mTOR signalling (although not unexpected), we find that the available knowledge, including that leucine supplementation can restore mTORC1 activity and improve muscle function in GAA KO mice and that you do not provide similar proof of concept of the potential efficacy of arginine supplementation in vivo, somewhat detract from the overall novelty of your findings.

For the above reasons, we are not persuaded that the manuscript provides the striking level of conceptual advance, novel mechanistic insight and/or the translational development we would like

to see in an EMBO Molecular Medicine article.

I am sorry that I could not bring better news.

Resubmission

23 August 2016

We submit for publication in EMBO Molecular Medicine, “**Modulation of mTOR signaling as a strategy for the treatment of Pompe disease**”, by Jeong-A Lim, Lishu Li, Orian Shirihai, Kyle M. Trudeau, Rosa Puertollano, and Nina Raben.

This manuscript is a much extended version of a previously submitted paper “Dissecting the mechanisms of mTOR dysregulation in Pompe disease” (EMM-2016-06547) which was rejected four months ago based on the lack of the “translational proof of concept”. In this new version we have added extensive data on the effects of the TSC2 inhibition in our knockout mouse model using AAV-mediated intramuscular injection of shRNA-TSC2. This genetic manipulation led to mTOR activation, reversal of atrophy, and a striking removal of the autophagic buildup – one of the major hurdles in enzyme replacement therapy - in the KO mice. Furthermore, a short-term dietary arginine supplementation activated mTOR signaling in the knockouts.

In our original cover letter, we predicted that the recent discovery of the lysosome as the site of mTORC1 activation and inactivation will profoundly affect the way we look at the lysosomal disorders and will be a big new development in the field. The evaluation of the mTORC1 status is particularly relevant to Pompe disease, a deficiency of lysosomal acid alpha-glucosidase (GAA), because it is a muscle wasting disorder, and mTORC1 is directly involved in the control of muscle mass. Profound muscle wasting persists and remains a major therapeutic challenge despite the currently available enzyme replacement therapy.

In this study, we have done a systematic analysis of mTOR pathway in Pompe muscle cells by evaluating mTOR activity, localization, regulation in response to nutrients, and its role in the control of protein synthesis and autophagy. We have found a profound mTOR signaling defect in the diseased muscle cells, suggested a model of mTOR dysregulation, and identified the sites of therapeutic intervention. We have employed lysosomal acidification and manipulated the TSC2-Rheb pathway (both in vitro and in vivo) to reverse the phenotype in Pompe cells.

Most exciting, by taking advantage of the recent data on the role of arginine in the regulation of mTOR signaling, we have shown that this amino acid reinstated mTOR activity in the diseased muscle cells and in the whole muscle of Pompe mice. This safe and effective treatment strategy may have broad relevance for a large group of neuromuscular and lysosomal disorders.

2nd Editorial Decision

26 September 2016

Thank you for the submission of your manuscript to EMBO Molecular Medicine. We have now heard back from two Reviewers whom we asked to evaluate your manuscript.

We are sorry that it has taken so long to get back to you on your manuscript. In fact, we experienced unusual difficulties in securing three willing and appropriate reviewers and in obtaining their evaluations in a timely manner. I am now proceeding based on the two available evaluations to avoid further delays.

As you will see, while reviewer 1 is quite positive, reviewer 2 is much more reserved and raises several issues of consequence on different aspects. I will not go into too much detail, as the evaluations are clear. In essence, one of the main points by reviewer 2 is that s/he is unconvinced that the mTOR phosphorylation pathway is affected based on the data presented. Reviewer 2 actually suggests that the retention of mTOR at the lysosome (and not the putative defective downstream phosphorylation) is the clearer phenotype and should be pursued directly as suggested in point 11. I should add that, during our cross-commenting exercise, reviewer 1 came to agree that retention of mTOR at the lysosome should be investigated further.

Reviewer 2 also notes a number of internal inconsistencies that need to be addressed. Finally, reviewer 1 is concerned that the statistical treatment of the data is of insufficient quality. This is a very important aspect for our title and I would encourage you to carefully read our guidelines concerning this (<http://embomolmed.embopress.org/authorguide#datapresentationformat>) and take action on the reviewer's points.

In conclusion, while publication of the paper cannot be considered at this stage, given the potential interest of your findings, we would be pleased to consider a revised submission, with the understanding that the reviewers' concerns must be addressed with additional experimental data where appropriate and that acceptance of the manuscript will entail a second round of review.

I look forward to seeing a revised form of your manuscript in due time.

***** Reviewer's comments *****

Referee #1 (Comments on Novelty/Model System):

This is a technically beautiful piece of work, which explains why skeletal muscle of patients with Pompe disease does not respond to enzyme replacement therapy. The problem in skeletal muscle is dysregulation of the mTOR pathway. Therefore restoration of the mTOR pathway could be therapeutic, especially when it can be achieved by administration of such a simple molecule as arginine. For the lysosomal disease field, this is very exciting news.

Referee #1 (Remarks):

This is an interesting paper which shows that dysregulation of the mTOR pathway is the major problem in skeletal muscle in Pompe disease. The following comments and suggestions are made in the interest of clarity. They are in the order of appearance in the text, not of significance.

1. Abstract: "the sites of therapeutic intervention" would be better as "potential sites for therapeutic intervention".
2. p. 3: The reference to the study by Shemesh comes off as sarcastic, and should be rephrased.
3. p. 4: The text speaks of a model which is immortalized GAA-deficient myoblasts, but in the next sentence states that it is myotubes, not myoblasts, that replicate the disease phenotype. That needs to be clarified.
4. On p. 7, the authors use concanamycin A, a specific inhibitor of the vacuolar V-ATPase. They call it "ConA", which is an abbreviation long used for the plant lectin concanavalin A. In fact, they get themselves confused, because in the legend to Fig. 6, they call the inhibitor "concanavalin A". Some other abbreviation would be preferable.
5. Statistics - the data are reported as mean {plus minus} SD for Figs 1-8, then switch to mean {plus minus} SE in Fig 9; fig 10 has both SD and SE. While using standard error and standard deviation are both valid ways to present data, it is confusing when both are used in the same paper, not to mention in the same figure. The explanation offered on p. 16, that SD was used for experiments in vitro and SE for experiments in vivo makes no sense, since statistics are independent of what is being analyzed. It seems to be a way of saying that different people were plotting their results in different ways. If that is the case, perhaps they could find a common way to plot their data. It is also not clear why the probability is presented only as <0.05. Just looking at the bar graphs and errors, it seems that some of the probabilities should be much smaller. Using the convention of 1 star for < 0.05, two stars for <0.01 and 3 stars for <0.001 would do justice to the very beautiful data on the gels.
6. p. 10 last paragraph. I find the discussion of other muscle disorders to be a distraction.
7. Fig 5D. Could the two bands of LC3 be labeled LC3I and LC3II?
8. Fig. 10 The effect of arginine treatment on the T37/46 phosphorylation site of panel E should also be analyzed quantitatively.
9. Fig S5. The finding of arginine activation of mTOR is presented as the most promising finding of this study. Why is some of the arginine data relegated to the Appendix?
10. EV figures. It is not clear where EV figures would go in the finished text.
11. EV Figure 1. Which cells?

Referee #2 (Comments on Novelty/Model System):

The model is adequate

Referee #2 (Remarks):

In the manuscript "Modulation of mTOR signaling as a strategy for the treatment of Pompe disease" by Jeong-A Lim et. al., the authors present a series of interesting studies, both in vitro and in vivo, directed towards the demonstration of a possible defect of the mTOR signaling pathway in the lysosomal storage disorder (LSD) Pompe disease. The manuscript constitutes an extensive analysis of many phosphorylation-mediated regulatory pathways of the signaling mechanisms that control mTOR activity. The authors also demonstrate the permanence of mTOR at the lysosomal membrane in enlarged lysosomes in cellular models of this LSD and propose several mechanisms to overcome the putative defects in mTOR activity.

Overall, although the methodologies employed are generally sound, and the manuscript contributes novel information including the observation that mTOR largely remains at the lysosomal membrane even under starvation conditions, and that the phenotype is only reverted by lysosomal acidification, the interpretation of the other sections of data not always matches the representative data displayed. In particular, the description of the results related to mTOR substrates phosphorylation not always correlates with that observed in the Western blots, and the data requires further re-interpretation.

Major comments

1. Although the authors propose that mTOR activity is defective, the high levels of phosphorylation observed for some of the mTOR substrates, seems to argue against an mTOR defect, but rather support an imbalance of the function of the substrates themselves. As an example, the phosphorylation levels of the translation inhibitor and mTOR substrate 4E-BP1 is upregulated in Pompe disease irrespective of the high levels of unphosphorylated protein present in KO cells (as observed in Figs 2A and 4A, for example). Also, the levels of phosphorylation of S6K appear marginally decreased and the levels of pS6 seem similar in WT and KO cells (Fig. 2F). Thus, it is clear from the WB that the differences between WT and KO cells are very mild and therefore they do not support a profound defect in mTOR activity in KO cells model of Pompe disease.
2. Further supporting a responsive mTOR pathway in KO cells, p-4E-BP1 levels appear to be responsive to starvation. In particular, in Figure 5 C, this reviewer observes that the protein is efficiently de-phosphorylated in response to AA and serum deprivation. In fact, a clear shift in the mobility of the protein in the electrophoresis analysis is manifested as the disappearance of a slower phosphorylated band with the appearance of 2 faster bands in both WT and KO cells in total protein WBs. The ratio of (p-4E-BP1 under ++ conditions/ p-4E-BP1 under - - conditions) should be then calculated, but in principle it looks similar for WT and KO.
3. Also, in Figure 5D, the analysis of macroautophagy requires further examination as the extremely subtle differences do not correlate with the text that describe the phenotype in KO cells as "failure to strongly induce autophagy". A thorough examination should include a full analysis of autophagic flux, p62 protein levels and long-live protein analysis under fed, starvation and CQ/BafA conditions.
4. For most Western blots, vinculin is included as a loading control. A problem that arises here is that vinculin consistently appears as a duplet in WT cells but as single band in KO cells. It is also not clear if or which of this bands is/are used for normalization and for quantitative analysis. This should be discussed and clarified and preferentially, a different loading control should be used.
5. Based on data showing increased phosphorylation of Akt (s473) but decreased phosphorylation of TSC2 (T1462) the authors concluded that suppression of mTORC1 activity is Akt-independent. This conclusion should be clarified and revisited as a failure of Akt to phosphorylate and inactivate TSC2 seems to be an Akt-dependent rather than independent mechanism.
6. Figure 3C. In order to support an AMPK-dependent mechanism of TSC2 activation the authors should measure Ser1345/Ser1387 phosphorylation. This antibody was used later in a different experiment but would help support their hypothesis and should be included in Figure 3C.
7. Fig. 3d: Is lysosomal TSC2 phosphorylated in either T1462 or Ser1387? This should be measured and included.
8. The levels of Akt phosphorylation, while elevated in vitro, seem normal in vivo (Fig. 4A). This should be discussed as the phenotype doesn't seem to fully recapitulate the in vitro phenotype.

9. Quantification in figure 4b is expressed as the ratio between the phospho-protein/the total protein for all proteins except AMPK. For uniformity, p-AMPK/AMPK should be presented. This is important since both the levels of phosphorylated and total protein seem elevated in the KO in the Western blot shown in 4A.

10. The model in Fig. 7 seems premature as this time as the interpretation that mTORC1 is not inactivated efficiently seem to contradict the data presented in Figure 5C.

11. The observation that mTOR is retained at the lysosomal membrane is very interesting and raises mechanistic questions. Is Rheb involved in mTOR retention? The authors show that overexpression of RheB increases mTOR activity. Does RheB overexpression also increase the accumulation of mTOR at the lysosomal membrane? What is the consequence of starvation in terms of mTOR activity under Rheb overexpression conditions? What is the status of Ragulator and Rag heterodimers and their role in mTOR retention in Pompe disease?

12. Many of the bar graphs require specification in the Figure legends in particular the number of samples used for statistical analysis seem to be missing in many of the figures and most of the Figure legends are written short-handed. Examples are: Figures 2E, 3B, 3E, 4E, 5B etc.

13. The significant differences between samples should be clearly specified. For example, in Figure 8D an asterisk was included in the last column (++) but it is not clear if this is different to all other groups in the figure. Multiple analyses between groups should be included here. Also, in 5E (LC3), is WT different from KO under HBSS conditions?

14. Arginine treatment is proposed to upregulate/normalize mTOR activity. Interestingly, among the many TSC2 functions that are independent of mTOR, regulation of amino acid transporters, and in particular non-canonical regulation of arginine uptake that is Rheb and TSC2-dependent but appears to be independent of mTOR has been described (EMBO Mol Med 2011 Apr; 3(4): 189-200.). Given the upregulated expression levels of TSC2 in KO cells this mechanism may have connotations in Pompe disease and should probably be discussed.

Minor comments

1. Confocal images should be color coded in the image to facilitate analysis
2. Please number Figures.

2nd Revision - authors' response

03 November 2016

We thank both reviewers for their useful comments and suggestions and we have revised the manuscript accordingly.

Referee #1 (Remarks):

This is an interesting paper which shows that dysregulation of the mTOR pathway is the major problem in skeletal muscle in Pompe disease. The following comments and suggestions are made in the interest of clarity. They are in the order of appearance in the text, not of significance.

We have addressed all the points raised by the reviewer and included the corrections in the revised manuscript.

1. *Abstract: "the sites of therapeutic intervention" would be better as "potential sites for therapeutic intervention".*

The word "potential" was added in the revised manuscript. The sentence in the abstract now reads: "Here, we report the dysregulation of mTOR signaling in the diseased muscle cells and we focus on potential sites for therapeutic intervention".

2. *p. 3: The reference to the study by Shemesh comes off as sarcastic, and should be rephrased.*

We rephrased the sentence which now reads: "The temptation to boost protein synthesis by stimulating the mTOR pathway is reflected in recent data showing that leucine supplementation halted the decline in muscle mass and reduced glycogen accumulation in GAA-KO muscle (Shemesh et al, 2014)".

3. p. 4: *The text speaks of a model which is immortalized GAA-deficient myoblasts, but in the next sentence states that it is myotubes, not myoblasts, that replicate the disease phenotype. That needs to be clarified.*

This point is clarified and the modified part reads as follows: “To explore the mTOR signaling pathway in Pompe disease, we took advantage of a recently developed *in vitro* model of the disease – GAA-deficient myotubes. These myotubes are formed from conditionally immortalized myoblasts derived from the GAA-KO mice; differentiated myotubes, but not myoblasts, contain large glycogen-filled lysosomes, thus replicating the disease phenotype (Spampanato et al, 2013)”.

4. *On p. 7, the authors use concanamycin A, a specific inhibitor of the vacuolar V-ATPase. They call it "ConA", which is an abbreviation long used for the plant lectin concanavalin A. In fact, they get themselves confused, because in the legend to Fig. 6, they call the inhibitor "concanavalin A". Some other abbreviation would be preferable.*

We have removed the abbreviation – thank you

5. *Statistics - the data are reported as mean \pm SD for Figs 1-8, then switch to mean {plus minus} SE in Fig 9; fig 10 has both SD and SE. While using standard error and standard deviation are both valid ways to present data, it is confusing when both are used in the same paper, not to mention in the same figure. The explanation offered on p. 16, that SD was used for experiments in vitro and SE for experiments in vivo makes no sense, since statistics are independent of what is being analyzed. It seems to be a way of saying that different people were plotting their results in different ways. If that is the case, perhaps they could find a common way to plot their data. It is also not clear why the probability is presented only as <0.05 . Just looking at the bar graphs and errors, it seems that some of the probabilities should be much smaller. Using the convention of 1 star for <0.05 , two stars for <0.01 and 3 stars for <0.001 would do justice to the very beautiful data on the gels.*

In the revised manuscript, all the data are reported as mean \pm SE and we use one star for P values <0.05 , two stars for P <0.01 , and three stars for P <0.001 . In addition we provide Source Data with actual P value for each test.

6. *p. 10 last paragraph. I find the discussion of other muscle disorders to be a distraction.*

We have considerably shortened this part of the discussion, but not eliminated it completely; we think it is reasonable to place the results in the context of other muscle disorders.

7. *Fig 5D. Could the two bands of LC3 be labeled LC3I and LC3II?*

The two bands are labeled as suggested.

8. *Fig. 10 The effect of arginine treatment on the T37/46 phosphorylation site of panel E should also be analyzed quantitatively.*

We did calculate the effect of arginine on T37/46 phosphorylation, but did not label the graph properly. This was corrected in the revised manuscript.

9. *Fig S5. The finding of arginine activation of mTOR is presented as the most promising finding of this study. Why is some of the arginine data relegated to the Appendix?*

The data are moved to the main text as Expanded view Figure EV5.

10. *EV figures. It is not clear where EV figures would go in the finished text.*

The EV figures are presented in the order they are mentioned in the text according to the guidelines.

11. *EV Figure 1. Which cells?*

These are WT and KO cells as indicated in the legend.

Referee #2 (Remarks):

In the manuscript "Modulation of mTOR signaling as a strategy for the treatment of Pompe disease" by Jeong-A Lim et. al., the authors present a series of interesting studies, both in vitro and in vivo, directed towards the demonstration of a possible defect of the mTOR signaling pathway in the lysosomal storage disorder (LSD) Pompe disease. The manuscript constitutes an extensive analysis of many phosphorylation-mediated regulatory pathways of the signaling mechanisms that control mTOR activity. The authors also demonstrate the permanence of mTOR at the lysosomal membrane in enlarge lysosomes in cellular models of this LSD and propose several mechanisms to overcome the putative defects in mTOR activity.

Overall, although the methodologies employed are generally sound, and the manuscript contributes novel information including the observation that mTOR largely remains at the lysosomal membrane even under starvation conditions, and that the phenotype is only reverted by lysosomal acidification, the interpretation of the other sections of data not always matches the representative data displayed. In particular, the description of the results related to mTOR substrates phosphorylation not always correlates with that observed in the Western blots, and the data requires further re-interpretation.

We understand the reviewer's concerns, and we have done additional experiments to clarify the mTOR status. We next detail our responses to each of the reviewer's comments.

Major comments

1. *Although the authors propose that mTOR activity is defective, the high levels of phosphorylation observed for some of the mTOR substrates, seems to argue against an mTOR defect, but rather support an imbalance of the function of the substrates themselves. As an example, the phosphorylation levels of the translation inhibitor and mTOR substrate 4E-BP1 is upregulated in Pompe disease irrespective of the high levels of unphosphorylated protein present in KO cells (as observed in Figs 2A and 4A, for example). Also, the levels of phosphorylation of S6K appear marginally decreased and the levels of pS6 seem similar in WT and KO cells (Fig. 2F). Thus, it is clear from the WB that the differences between WT and KO cells are very mild and therefore they do not support a profound defect in mTOR activity in KO cells model of Pompe disease.*

The increase in both phosphorylated p-4E-BP1^{T37/46} and total 4E-BP1 is invariably seen in Pompe myotubes and whole muscle from GAA-KO mice (the data in whole muscle is consistent with our previously reported findings). This increase in both forms of 4EB-P1 presented a conundrum, and in the manuscript we did acknowledge the problem of assessing mTOR activity. That is why we evaluated the level of non-phosphorylated 4E-BP1 and did immunoprecipitation experiments to demonstrate an increase in eIF4E/4E-BP1 binding in KO cells to account for the decrease in protein synthesis.

In general, in several experiments we encountered a problem of increased levels of both phosphorylated and total forms of a protein (as, for example, in case of AMPK in addition to 4E-BP1). We reasoned that at least in some of these cases the ratio of phosphorylated/total may be misleading; instead, we chose to put emphasis on the level of active form of the protein, namely the level of non-phosphorylated 4E-BP1 and phosphorylated AMPK.

However, we agree with the reviewer on the importance of further evaluating the mTOR status. To better monitor mTOR activity in Pompe muscle cells we have added experiments using the anti p-4EBP1 Ser65 Ab (this Ab was used in different experiments, namely in arginine experiments). Although phosphorylation of 4E-BP1 on T37/46 is commonly used to assess mTOR activity, phosphorylation at S65 is a more reliable indicator since it is serum-responsive and rapamycin-sensitive, whereas phosphorylation T37/46 is only partially sensitive to these treatments (Gingras et al, 2001). Using these antibody, we again found an increase in both forms, but the ratio was significantly decreased in Pompe cells. These data and the reference are included in the revised manuscript. In addition, we re-structured this part of the paper to first present the data which may explain the reduction of protein translation followed by the data on mTOR activity.

As for the levels of phosphorylation of S6K and S6, the difference between WT and KO is, indeed, not huge, but significant. **Five** independent experiment were done for S6K and **six** for S6 (figure 2). Taken together, we think the data justify diminished mTOR activity, but we follow the reviewer's suggestion and refrain from using "a profound defect in mTOR activity".

As for the reviewer's suggestion of "an imbalance of the function of the substrates themselves", we did discuss this possibility in the legend for Figure EV1 of the original manuscript. In the revised paper, we moved this part to the main text (Figure 5).

2. Further supporting a responsive mTOR pathway in KO cells, p-4E-BP1 levels appear to be responsive to starvation. In particular, in Figure 5 C, this reviewer observes that the protein is efficiently de-phosphorylated in response to AA and serum deprivation. In fact, a clear shift in the mobility of the protein in the electrophoresis analysis is manifested as the disappearance of a slower phosphorylated band with the appearance of 2 faster bands in both WT and KO cells in total protein WBs. The ratio of (p-4E-BP1 under ++ conditions/ p-4E-BP1 under - - conditions) should be then calculated, but in principle it looks similar for WT and KO.

Indeed, there is a shift down in the mobility of the protein in both WT and KO, but the slower phosphorylated band is much more prominent in KO compared to WT. All we are saying here is that the degree of dephosphorylation in the KO was less pronounced, particularly when the cells were treated with medium lacking amino acids only. We clarified this point in the revised manuscript and we added a graph to illustrate it (Fig 5 B). In addition, we repeated the experiments with p-4E-BP1 Ser65 and observed a similar pattern (Fig. 5B), again suggesting a defective dephosphorylation of the mTOR substrate.

3. Also, in Figure 5D, the analysis of macroautophagy requires further examination as the extremely subtle differences do not correlate with the text that describe the phenotype in KO cells as "failure to strongly induce autophagy". A thorough examination should include a full analysis of autophagic flux, p62 protein levels and long-live protein analysis under fed, starvation and CQ/BafA conditions.

Over the course of the years, we have done extensive analysis of autophagy in KO myotubes and whole muscle of the GAA-KO mice. The in vivo studies included analysis of LC3 transgenics on Pompe background, analysis of muscle-specific autophagy deficient Pompe mice, examination of live fibers expressing LAMP and LC3, etc. These studies revealed a severe impairment of autophagic pathway in Pompe muscle and established autophagic buildup as a prominent pathology in the diseased muscle.

In KO myotubes we analyzed autophagy by overexpression of RFP-GFP-LC3; by evaluating the levels of p62 and ubiquitinated proteins; by LC3 levels after CQ or bafilomycin treatment. These studies showed that the changes in KO myotubes are much more subtle: increase in LC3-II/total ratio, partial autophagic block and absence of autophagic buildup.

All these data have already been published. In our recent paper (Lim et al, 2015) we state that **"The levels of LC3-II in the presence of chloroquine or bafilomycin A₁ (another lysosomal inhibitor) increased in both WT and KO cells but the effect was less pronounced in the KO myotubes"**.

The point we would like to make in this part of the paper is that the autophagic response to starvation in KO is not as strong as in the WT. To strengthen this, as suggested by the reviewer, we treated WT and KO myotubes with CQ/starvation; immunostaining with LC3, again, showed lesser autophagy response in KO. We have added these results in the revised manuscript (Fig EV2), and we indicate in the discussion that the high basal level of autophagy may contribute to weakened autophagy response. In general, we think that further evaluation of autophagy in KO myotubes may distract from the data on TSC2-mediated removal of autophagic buildup in vivo.

4. For most Western blots, vinculin is included as a loading control. A problem that arises here is that vinculin consistently appears as a duplet in WT cells but as single band in KO cells. It is also not clear if or which of this bands is/are used for normalization and for quantitative analysis. This should be discussed and clarified and preferentially, a different loading control should be used.

Vinculin and its splice variant, metavinculin, are commonly seen in both WT and KO cells, although the ratio of these forms is different. Over the years, we have used tubulin, GAPDH, and vinculin as loading controls, and have not detected any difference. We provide an additional file for the reviewer (**figure 1**) showing the comparison between vinculin and GAPDH. For normalization and for quantitative analysis we use both bands, and we clarify this point in the revised manuscript (Legend for Figure 2).

5. Based on data showing increased phosphorylation of Akt (s473) but decreased phosphorylation of TSC2 (T1462) the authors concluded that suppression of mTORC1 activity is Akt-independent. This conclusion should be clarified and revisited as a failure of Akt to phosphorylate and inactivate TSC2 seems to be an Akt-dependent rather than independent mechanism.

We do not make any conclusion about mTOR activity based on increased phosphorylation of Akt (S473) but decreased phosphorylation of TSC2 (T1462). Our conclusion is very much the same as suggested by the reviewer: “Despite this increase, AKT-mediated phosphorylation of TSC2^{T1426} was decreased in KO cells, suggesting a failure of AKT to inhibit TSC2.” However, we agree that the sentence that follows - “Thus, the decreased mTORC1 activity in the diseased cells appears to be AKT-independent” – may be a cause of confusion. In the revised manuscript, we removed it and discuss the mTOR activity elsewhere.

6. Figure 3C. In order to support an AMPK-dependent mechanism of TSC2 activation the authors should measure Ser1345/Ser1387 phosphorylation. This antibody was used later in a different experiment but would help support their hypothesis and should be included in Figure 3C.

As suggested by the reviewer, we evaluated the levels of phosphorylated AMPK^{T172} and p-TSC2^{S1387}. These experiments were done using lysosomal fraction since recent data indicate that AMPK-mediated activation of TSC2^{S1387} leads to lysosomal recruitment of TSC2. We found an increase in both in the KO, thus supporting our hypothesis that activation of AMPK-TSC2 pathway leads to diminished basal mTOR activity; these data are included in the revised manuscript (Figure 3).

We realized that the original Figure 3C showing AMPK response to starvation does not belong to the set of data presented in Figure 3; the basal levels of total and p-AMPK are shown in Figure 3A, and the effect of starvation is discussed later (Figures 5 and 6 in the revised manuscript).

7. Fig. 3d: Is lysosomal TSC2 phosphorylated in either T1462 or Ser1387? This should be measured and included.

Total TSC2 is shown in Fig. 3D. We did measure AKT-mediated p-TSC2^{T1462} form in the lysosomal fraction and did not detect a clear band, again supporting our hypothesis. These data, as well as the measurements of TSC2^{Ser1387} in the lysosomal fraction are added in the revised manuscript (as indicated in our response to the point 6).

8. The levels of Akt phosphorylation, while elevated in vitro, seem normal in vivo (Fig. 4A). This should be discussed as the phenotype doesn't seem to fully recapitulate the in vitro phenotype.

In the original manuscript, we indicate that “the phosphorylation levels of AKT (p-AKT^{S473}) were similar in WT and KO cells on day 4-5 in differentiation medium (not shown), and were even increased in the KO at a later stage of myotubes differentiation (Fig 3A).” The in vivo experiments evaluating AKT levels were repeated multiple times, and we observe the same pattern as in vitro – no change or a slight increase (in the additional file for the reviewer we provide these extra data; **figure 2**). Therefore, we prefer not to make a clear distinction between the in vitro and in vivo phenotypes, particularly because the emphasis here is that there is no decrease in the levels of p-AKT to account for the reduced mTOR activity.

9. Quantification in figure 4b is expressed as the ratio between the phospho-protein/the total protein for all proteins except AMPK. For uniformity, p-AMPK/AMPK should be presented. This is

important since both the levels of phosphorylated and total protein seem elevated in the KO in the Western blot shown in 4A.

Please see our response to point #1.

10. The model in Fig. 7 seems premature as this time as the interpretation that mTORC1 is not inactivated efficiently seem to contradict the data presented in Figure 5C.

We would very much prefer to keep the model, particularly because it reflects the retention of mTOR at the lysosome, a finding that is not questioned by the reviewer. Furthermore, in the revised manuscript we provide additional data to strengthen the results, and we call it “a putative model”.

11. The observation that mTOR is retained at the lysosomal membrane is very interesting and raises mechanistic questions. Is Rheb involved in mTOR retention? The authors show that overexpression of RheB increases mTOR activity. Does RheB overexpression also increase the accumulation of mTOR at the lysosomal membrane? What is the consequence of starvation in terms of mTOR activity under Rheb overexpression conditions? What is the status of Ragulator and Rag heterodimers and their role in mTOR retention in Pompe disease?

The mechanism of Rheb interaction and activation of mTOR is still not completely understood, and this issue is beyond the scope of this study. Based on the published data, we did not expect the involvement of Rheb in the recruitment of mTOR to the lysosomes. What we expected was a decrease in Rheb activity because of the excess of TSC2 at the lysosomal membrane. We have tried to measure Rheb activity by using the configuration-specific antibody that recognizes and immunoprecipitates Rheb-GTP to pull down the active Rheb, followed by immunoblot analysis with anti-Rheb rabbit polyclonal antibody (currently there is no direct assay to measure the activity of Rheb GTPase). Unfortunately, we were not convinced by the results (these data are shown in the additional file for the reviewer; **figure 3**). We have also tried unsuccessfully immunostaining with Rheb antibody.

In the revised manuscript we expanded the data with Rheb overexpression and we found the same pattern of mTOR accumulation at the lysosome as in non-treated cells: mTOR remains at the lysosome after starvation in the KO but not in the WT. These data are included in the revised manuscript (figure EV2).

As for the Rag heterodimers, the evaluation of their nucleotide state status is far from trivial, particularly if one is interested in the activity of endogenous Rags. However, to address the reviewer’s comment, we tried to evaluate the localization of folliculin that activates RagC/D in Rag heterodimers. As shown in the literature, folliculin localizes at the lysosome during amino-acid starvation and leaves the lysosomal site upon amino acid stimulation; in other words, folliculin and mTOR move into opposite directions, and the localization of folliculin may serve as an indirect indicator of the nucleotide state of Rag heterodimers. Unfortunately, the available antibodies that recognize mouse folliculin did not work in muscle cells. We have tried both immunostaining and western blot; no right size band was detected by western (please see **figure 4** in the additional file for the reviewer).

12. Many of the bar graphs require specification in the Figure legends in particular the number of samples used for statistical analysis seem to be missing in many of the figures and most of the Figure legends are written short-handed. Examples are: Figures 2E, 3B, 3E, 4E, 5B etc.

In the revised manuscript, all the data are reported as mean \pm SE; we use one star for < 0.05 , two stars for < 0.01 , and three stars for < 0.001 . We have also expanded the legends to include additional information and the number of experiments. We provide Source Data with actual P value for each test.

13. The significant differences between samples should be clearly specified. For example, in Figure 8D an asterisk was included in the last column (++) but it is not clear if this is different to all other

groups in the figure. Multiple analyses between groups should be included here. Also, in 5E (LC3), is WT different from KO under HBSS conditions?

The significant differences are clearly specified in the revised manuscript.

14. Arginine treatment is proposed to upregulate/normalize mTOR activity. Interestingly, among the many TSC2 functions that are independent of mTOR, regulation of amino acid transporters, and in particular non-canonical regulation of arginine uptake that is Rheb and TSC2-dependent but appears to be independent of mTOR has been described (EMBO Mol Med 2011 Apr; 3(4): 189-200.). Given the upregulated expression levels of TSC2 in KO cells this mechanism may have connotations in Pompe disease and should probably be discussed.

Thank you for pointing to this review paper summarizing the involvement of Rheb in arginine uptake in different yeast strains and in Drosophila S2 cell. However, the role of mTOR in this TSC-Rheb-mediated process is still unclear. Arginine trafficking in Pompe muscle cells and in whole muscle is, no doubt, an interesting question, but it is beyond the scope of this study. Therefore, we prefer to avoid discussing the subject.

Minor comments

1. Confocal images should be color coded in the image to facilitate analysis

The images are now color coded

2. Please number Figures.

The numbers are now embedded in the Figures.

References

Gingras AC, Raught B, Gygi SP, Niedzwiecka A, Miron M, Burley SK, Polakiewicz RD, Wyslouch-Cieszyńska A, Aebersold R, Sonenberg N (2001) Hierarchical phosphorylation of the translation inhibitor 4E-BP1. *Genes Dev* **15**: 2852-2864

Lim JA, Li L, Kakhlon O, Myerowitz R, Raben N (2015) Defects in calcium homeostasis and mitochondria can be reversed in Pompe disease. *Autophagy* **11**: 385-402

Shemesh A, Wang Y, Yang Y, Yang GS, Johnson DE, Backer JM, Pessin JE, Zong H (2014) Suppression of mTOR1 Activation in Acid- α -Glucosidase Deficient Cells and Mice is Ameliorated by Leucine Supplementation. *American journal of physiology Regulatory, integrative and comparative physiology*

Spampanato C, Feeney E, Li L, Cardone M, Lim JA, Annunziata F, Zare H, Polishchuk R, Puertollano R, Parenti G, Ballabio A, Raben N (2013) Transcription factor EB (TFEB) is a new therapeutic target for Pompe disease. *EMBO Mol Med* **5**: 691-706

3rd Editorial Decision

06 December 2016

Thank you for the submission of your revised manuscript to EMBO Molecular Medicine. We have now received the enclosed reports from the referees that were asked to re-assess it. As you will see the reviewers are now globally supportive and I am pleased to inform you that we will be able to accept your manuscript pending the following.

While performing our pre-publishing quality control and image screening routines, we noticed issues pertaining to a number figures, as listed below:

1) The blots appear to be presented with an excessive contrast setting. This does not allow us to properly carry out our image analysis and also introduces artifacts, which may be later construed to suggest manipulation. Please provide lesser-contrasted more faithful to the original figures.

2) We have detected a number of possible instances suggestive of splicing-in of lanes in the blots. Please be reminded that "Images gathered at different times or from different locations should not be combined into a single image, unless it is stated that the resultant image is a product of time-averaged data or a time-lapse sequence. If juxtaposing images is essential, the borders should be clearly demarcated in the figure and described in the legend" (<http://embomolmed.embopress.org/authorguide>). Please provide amended images and an explanation of the occurrences together with the source data (see also below).

Please submit your revised manuscript within two weeks. I look forward to seeing a revised form of your manuscript as soon as possible.

***** Reviewer's comments *****

Referee #1 (Comments on Novelty/Model System):

This paper proposes dysregulation of mTOR signaling as the major metabolic problem underlying Pompe disease in muscle, explaining why the disorder does not respond to enzyme replacement therapy. It proposes a therapy based on restoring the mTOR pathway. The data are convincing, and the authors have responded to all my comments on the previous version.

Referee #1 (Remarks):

This paper contains an enormous amount of information, and even though the data are clearly presented, it may still be somewhat difficult for a non-specialist. In spite of that, I am very enthusiastic about this manuscript.

Referee #2 (Remarks):

The authors have been responsive and have included additional data and analysis to improve the manuscript.

3rd Revision - authors' response

12 December 2016

Here we resubmit the revised manuscript "**Modulation of mTOR signaling as a strategy for the treatment of Pompe disease**" (EMM-2016-06547-Q).

We thank both reviewers for evaluating the revised manuscript and for their support.

As requested, we have introduced the following changes:

1. The blots with excessive contrast settings are now presented as less-contrasted figures.
2. We provide PDF file ("Source Data") that contains the original, uncropped and unprocessed scans of all the gels used in the manuscript. The statement "Source data are available online for this figure" is added to the corresponding Figure legends.
3. Figure S5 in the Appendix is a montage of confocal images of fibers from the infected muscle of a GAA-KO mouse. We now state it clearly in the legend.
4. A running title is included.
5. "The Paper Explained" section is incorporated in the main text.
6. The name of the statistical test and the number of independent experiments for each graph are included in the Legends.
7. The list of actual *P*-values for each test is provided in Table EV1.

We hope the revised manuscript will be suitable for publication in the journal.

3rd Editorial Decision

14 December 2016

Thank you for the submission of your revised manuscript to EMBO Molecular Medicine. Most of my editorial requests have been now dealt with. However important issues remain that prevent us from moving forward with your manuscript.

In my previous decision letter I had specifically mentioned, that while performing our pre-publishing quality control and image screening routines, we had detected a number of possible instances suggestive of splicing-in/out of lanes in the blots. I had also drawn your attention to the fact that as per our guidelines (and indeed as good practice dictates): "Images gathered at different times or from different locations should not be combined into a single image, unless it is stated that the resultant image is a product of time-averaged data or a time-lapse sequence. If juxtaposing images is essential, the borders should be clearly demarcated in the figure and described in the legend". I had asked you therefore to provide amended images and an explanation of the occurrences together with the source data (see also below).

I note that you have provided the source data but you neither challenged nor explained these possible splicing occurrences (but for fig. S5).

Upon further analysis and thanks to the source data you provided, we take note that Fig 5A and D and 9F, which we had previously flagged, are indeed composite images with (rather carefully crafted) splicing out of central lanes vs. the original gel.

Given the source data provided, we are obviously not questioning the legitimacy of the data at this stage, but I must reiterate that this mode of presentation cannot be accepted and must be remedied with appropriately revised figures in which the ex-post juxtaposed blots must be separated by either a clear black line or white space, and explained in the legends. I hope you realize that this is especially in your best interest, to avoid embarrassing and potentially harmful challenges on the integrity of your data once the manuscript is published.

I look forward to your next final version as soon as possible so that we can rapidly proceed with acceptance.

4th Revision - authors' response

15 December 2016

Here we resubmit the corrected manuscript "**Modulation of mTOR signaling as a strategy for the treatment of Pompe disease**" (EMM-2016-06547-Q).

I apologize for the problem. It was my understanding that if the samples were run on the same gel (which is our case), the lanes can be removed without demarcating the lines. We have corrected Fig 5A and D and 9F, and inserted lines at the appropriate places. We now state in the legends that these are composite images. Furthermore, in the Source data we provide an explanation of why the lanes were removed.

We hope the manuscript will be (finally) suitable for publication in the journal.

Once again, sorry for my mistake.

Corresponding Author Name: Nina Raben
Journal Submitted to: EMBO Mol Med
Manuscript Number: EMM-2016-06547-Q-V3